# Orphan receptor GPR158 controls stress-induced depression

Laurie P Sutton[1†], Cesare Orlandi[1†], Chenghui Song[1], Won Chan Oh[2],
Brian S Muntean[1], Keqiang Xie[1], Alice Filippini[3], Xiangyang Xie[1],
Rachel Satterfield[2], Jazmine D W Yaeger[4,5], Kenneth J Renner[4,5],
Samuel M Young Jr[2,6,7,8], Baoji Xu[1], Hyungbae Kwon[2,9], Kirill A Martemyanov[1*]

[1]Department of Neuroscience, The Scripps Research Institute, Jupiter, United
States; [2]Max Planck Florida Institute for Neuroscience, Jupiter, United States;
[3]Department of Molecular and Translational Medicine, University of Brescia, Brescia,
Italy; [4]Center for Brain and Behavior Research, University of South Dakota,
Vermillion, United States; [5]Department of Biology, University of South Dakota,
Vermillion, United States; [6]Department of Anatomy and Cell Biology, University of
Iowa, Iowa, United States; [7]Aging Mind and Brain Initiative, University of Iowa,
Iowa, United States; [8]Department of Otolaryngology, Carver College of Medicine,
University of Iowa, Iowa, United States; [9]Max Planck Institute of Neurobiology,
Martinsried, Germany

*For correspondence:
kirill@scripps.edu

†These authors contributed
equally to this work

Competing interests: The
authors declare that no
competing interests exist.

Reviewing editor: Richard D
Palmiter, Howard Hughes
Medical Institute, University of
Washington, United States

**Abstract** Stress can be a motivational force for decisive action and adapting to novel
environment; whereas, exposure to chronic stress contributes to the development of depression
and anxiety. However, the molecular mechanisms underlying stress-responsive behaviors are not
fully understood. Here, we identified the orphan receptor GPR158 as a novel regulator operating in
the prefrontal cortex (PFC) that links chronic stress to depression. GPR158 is highly upregulated in
the PFC of human subjects with major depressive disorder. Exposure of mice to chronic stress also
increased GPR158 protein levels in the PFC in a glucocorticoid-dependent manner. Viral
overexpression of GPR158 in the PFC induced depressive-like behaviors. In contrast GPR158
ablation, led to a prominent antidepressant-like phenotype and stress resiliency. We found that
GPR158 exerts its effects via modulating synaptic strength altering AMPA receptor activity. Taken
together, our findings identify a new player in mood regulation and introduce a pharmacological
target for managing depression.
DOI: https://doi.org/10.7554/eLife.33273.001

## Introduction

An individual's emotional response to stress is determined by genetic and environmental factors.
While most reactions to stressful events trigger adaptive changes to maintain normal psychological
and physical functioning, their derailment could produce maladaptive behavioral responses leading
to the development of major depressive disorders (MDD) (*Mayberg et al., 1999*). It has been well
recognized that medial prefrontal cortex (mPFC) plays a key role in controlling mood regulation.
Changes in the morphology and/or function of mPFC neurons and local circuitry are well docu-
mented to contribute to the development of MDD (*Drevets et al., 1997*; *Rajkowska et al., 1999*).
Studies in animal models indicate that loss of mPFC control drives anxiety- and depressive-like phe-
notypes (*Amat et al., 2005*; *Shrestha et al., 2015*), while treatments that act within the mPFC pro-
mote a therapeutic effect (*Kumar et al., 2013*). The molecular mechanisms underlying the pathology
of MDD are complex and not fully understood but most models agree that they are related to

abnormal processing of neurotransmitter signals most notably by the monoamine, GABA and gluta-mate systems (*Hamon and Blier, 2013*). It is thought that these changes occur as a result of mal-adaptive plasticity in response to several biological and environmental factors and that understanding the mechanisms behind this connection would be a prerequisite for the development of new therapeutic or prophylactic interventions.

Exposure to chronic stress is among the most powerful insulting factors that exerts significant negative effects on mood (*Lucassen et al., 2014*). Exposure to stressful events is thought to underlie long-term neuroadaptations observed in MDD including dendritic remodeling, spine loss, and altered synaptic transmission (*Arnsten, 2015*; *Radley et al., 2006*). Much of these effects are medi-ated by the glucocorticoids whose unrestraint release during chronic stress often triggers develop-ment of maladaptive responses (*Myers et al., 2014*). Glucocorticoids engage steroid receptors to initiate transcription of responsive genes responsible for lasting alterations in cellular function well beyond the time period of stress exposure (*Kadmiel and Cidlowski, 2013*). However, the roles of stress-induced genes in mediating cellular changes in mPFC and driving maladaptive behavioral phe-notypes are not well understood.

Several GPCRs have been implicated in the pathophysiology of MDD, as well as the effects of stress and pharmacological modulation of monoamine GPCRs is a mainstay practice in treating MDD (*Hamon and Blier, 2013*; *Tomita et al., 2013*). Still, therapeutic efficacy of available medications is limited, which is largely attributed to insufficient understanding of signaling cascades and neuronal circuits involved in modulating mood and stress responses, making identification of new players an attractive goal. In this study, we tap into the uncharted biology of orphan GPCR-like receptors screening for the novel stress-regulated genes in the mPFC. We identify the orphan receptor GPR158 as a key modulator of stress-induced depression. We show involvement of GPR158 in pathology of MDD in humans and demonstrate that manipulation with GPR158 in animal models produces antidepressive-like behaviors and stress resilience.

## Results

### Identification of GPR158 as a stress-inducible receptor modulated by glucocorticoids in the PFC

With the intent of identifying novel players involved in the regulation of stress-mediated responses we focused our attention on orphan receptors, increasingly recognized for their potential (*Civelli et al., 2013*). We first defined the landscape of orphan receptors expressed in the brain by performing next generation sequencing of coding mRNAs, revealing that many orphan receptors are abundantly expressed in the mouse brain (*Figure 1A*). Subsequent proteomic profiling con-firmed the expression of many of them specifically in the PFC (*Figure 1B*). We then analyzed the effect of chronic physical restraint stress (PRS) on the protein levels of several candidates from this list in the mPFC. Mice subjected to PRS revealed a significant upregulation in the levels of GPR158 (*Figure 1C*), coincidently amongst the most prominent GPCRs (*Figure 1—figure supplement 1A* and *Supplementary file 1*) and, by far, the most abundant orphan receptor in the PFC (*Figure 1B*). PRS exposure did not affect GPR158 levels in other brain regions including the hippocampus, cau-date-putamen or cerebellum (*Figure 1—figure supplement 1B*). High-resolution in situ hybridization confirmed high expression levels of *Gpr158* message in the mPFC (*Figure 1D*). To evaluate the cell-type specific *Gpr158* expression pattern, we employed high-resolution double in situ hybridization with specific markers. We found *Gpr158* to be expressed in the majority of glutamatergic and in sub-populations of GABAergic neurons but largely absent in glia (*Figure 1—figure supplement 2A–D*).

Interestingly, in vitro studies implicated this receptor in the modulation of cellular signaling (*Orlandi et al., 2012*; *Patel et al., 2013*), while its close homolog, GPR179 is an established modula-tor of synaptic signaling in the retina (*Audo et al., 2012*; *Peachey et al., 2012*) making it an interest-ing candidate to pursue. In support of an involvement of chronic stress in controlling GPR158 levels in the mPFC, we observed that a second stress paradigm, unpredictable chronic mild stress (UCMS), and an independent cohort of mice subjected to PRS showed, again, a significant upregulation of GPR158 (*Figure 1E*). Furthermore, a positive correlation between GPR158 levels in the mPFC and immobility in the tail suspension test was found in both models pointing to the involvement of GPR158 in stress behaviors (*Figure 1F*). In contrast, we found that mice subjected to acute PRS did

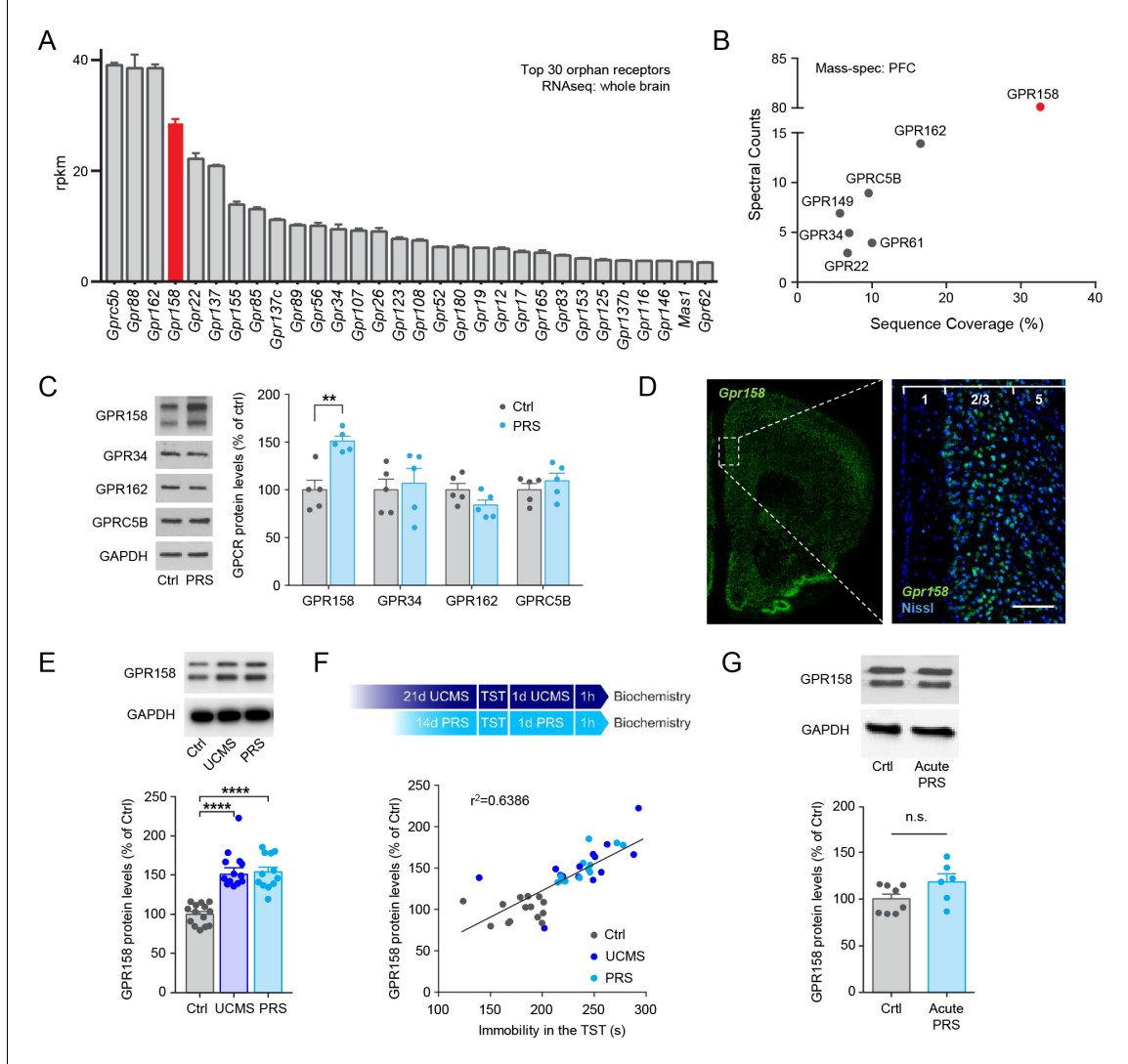

**Figure 1.** Stress upregulates GPR158 expression in the mPFC. (**A**) RNA sequencing results from the whole mouse brain showing the top 30 orphan receptors. (**B**) Mass spectrometry analysis of the orphan receptors expressed in the PFC of mice. (**C**) Representative western blots and protein level quantification of several orphan receptors identified in the mPFC of control non-stressed (Crtl) and mice subjected to chronic Physical Restraint Stress (PRS) (n = 4–5 mice/group; Student's t test; *p<0.05). (**D**) In situ hybridization showing the expression of GPR158 in the mPFC of mouse brain. (**E**) Mice subjected to PRS and unpredictable chronic mild stress (UCMS) show elevated GPR158 protein levels in the mPFC (n = 12–14 mice/group, one-way ANOVA with Bonferroni *post hoc* test). (**F**) Top panel shows scheme of stress paradigms. Graph showing a significant correlation between GPR158 protein levels and immobility in the TST of mice subjected to chronic stress (Pearson $R^2$ = 0.6386, p<0.0001). (**G**) Mice subjected to 1 day of physical resistant stress (Acute PRS, 3 hr, n = 6) show no difference in GPR158 protein levels compared to control non-stressed mice (Crtl, n = 8) in the mPFC.

DOI: https://doi.org/10.7554/eLife.33273.002

The following figure supplements are available for figure 1:

**Figure supplement 1.** GPR158 is among the most expressed GPCR in the PFC.

DOI: https://doi.org/10.7554/eLife.33273.003

**Figure supplement 2.** GPR158 expression in several neuronal populations.

DOI: https://doi.org/10.7554/eLife.33273.004

not show a significant up-regulation of GPR158 protein levels in the mPFC (*Figure 1G*). Using immunohistochemistry to distinguish glutamatergic and GABAergic populations in combination with in situ hybridization using a probe against *Gpr158*, we quantified *Gpr158* mRNA levels in layer 2/3 of the mPFC. This approach revealed that PRS exposure selectively up-regulated *Gpr158* in the glutamatergic neurons but not in GABAergic neurons suggesting cell-specific modulation of GPR158-

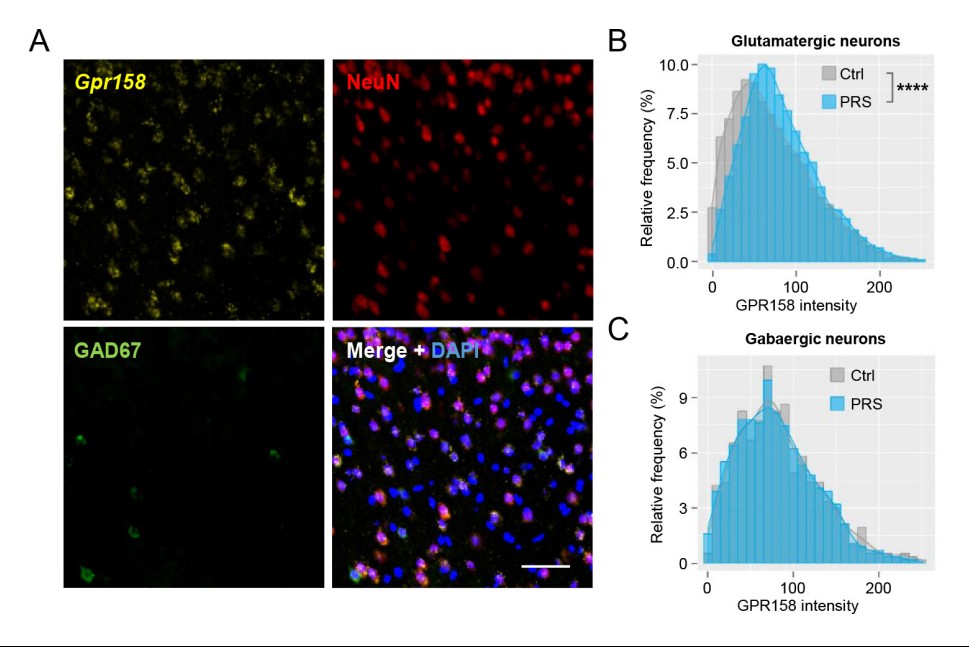

**Figure 2.** Chronic stress regulates GPR158 levels in glutamatergic neurons of the layer 2/3 of the mPFC. (**A**) Representative image of in situ hybridization using a probe against GPR158 (yellow) and co-immunostaining with antibodies against NeuN (red) and GAD67 (green) in the layer 2/3 of mPFC (nuclei stained with DAPI in blue, scale bar = 50 μm). (**B**) Frequency distribution of GPR158 mRNA levels in glutamatergic neurons in the mPFC layer 2/3 (Ctrl n = 5 mice/6486 neurons; PRS n = 5 mice/6400 neurons; bin width = 10; p<0.0001 Kolmogorov–Smirnov test). (**C**) Frequency distribution of GPR158 expression in GAD67- positive neurons (Ctrl n = 5 mice/569 cells; PRS n = 5 mice/564 cells; bin width = 10; p=0.9599 Kolmogorov-Smirnov test).
DOI: https://doi.org/10.7554/eLife.33273.005

mediated effects (*Figure 2A–C*). Altogether we show that chronic stress induces overexpression of the orphan receptor GPR158 in glutamatergic neurons of the mPFC.

We next examined the effects of glucocorticoids on GPR158 levels, as they mediate the effects of chronic stress on gene expression (*Al-Safadi et al., 2015*; *Gray et al., 2014*). Chronic corticosterone administration elevated GPR158 protein expression in the mPFC and GPR158 levels correlated with the immobility time in the TST (*Figure 3A*). In contrast, acute injection of corticosterone did not affect GPR158 protein levels (*Figure 3B*). Furthermore, a 7 day treatment of primary cortical neurons with the glucocorticoid receptor agonist dexamethasone upregulated GPR158 expression (*Figure 3C*). Finally, we found that systemic administration of the glucocorticoid receptor antagonist, RU-486 blocked the increase in GPR158 expression induced by chronic stress in the mPFC (*Figure 3D*). Taken together with the presence of glucocorticoid responsive elements in the promoter of GPR158 gene and its regulation by glucocorticoids in peripheral cells (*Patel et al., 2013*), these results suggest that chronic stress influences GPR158 protein levels via corticosteroid-induced regulation of GPR158 gene expression.

## GPR158 overexpression in mouse mPFC is sufficient to induce a depressive-like phenotype

To investigate the role of increased GPR158 levels in depressive-like behaviors affected by stress, we utilized a viral approach to overexpress GPR158 in the mPFC (*Figure 4A and B*). Bilateral injections of GPR158-expressing viral vector (GPR158-Ad) but not control virus (EGFP-Ad) significantly elevated GPR158 expression in the region (*Figure 4C*). Strikingly, we found that mice given GPR158-Ad infusions displayed increased immobility in the tail suspension and forced swim tests while their behavior in the marble burying and elevated plus maze tests was unchanged relative to control group treated with EGFP-Ad (*Figure 4D–G*). To assess the overall consistency along similar behavioral dimensions and reduce the bias introduced by individual tests that probe different

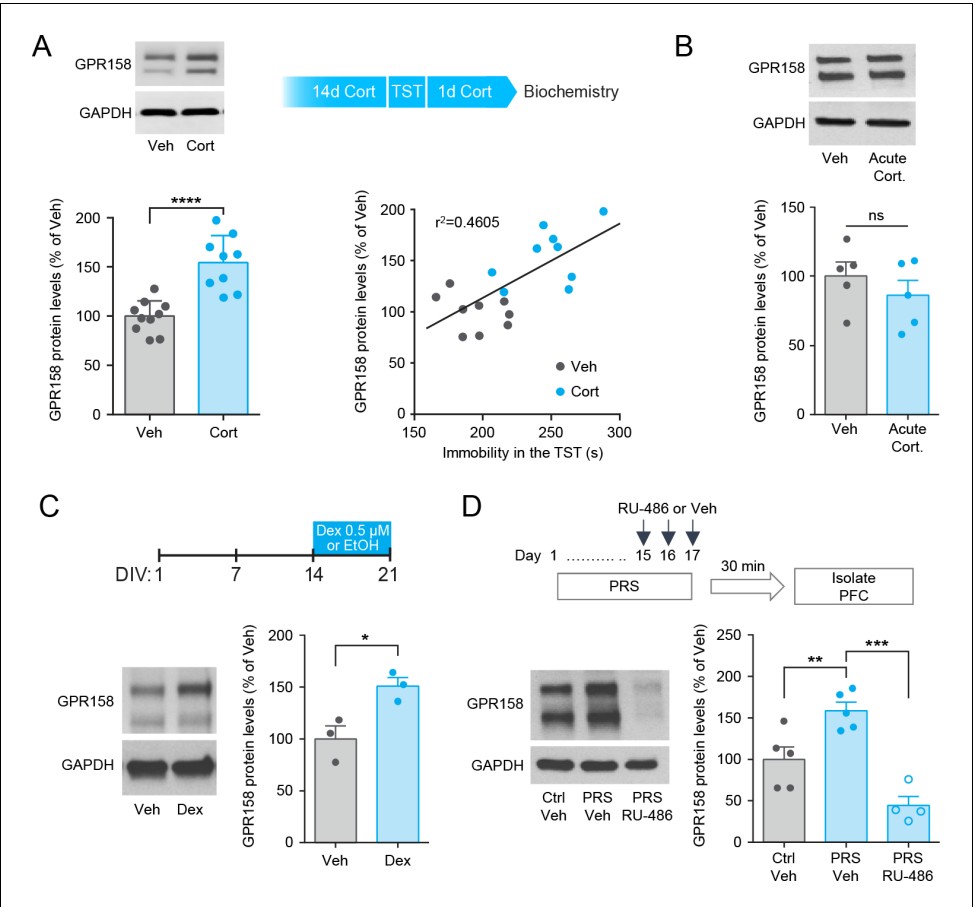

**Figure 3.** GPR158 is increased in the mouse mPFC by a corticosterone-mediated mechanism. (**A**) GPR158 protein levels are increased in the mPFC of mice chronically treated with corticosterone (n = 9–10 mice/group, Student's *t* test). Top panel depicts corticosterone paradigm. Mice treated with corticosterone exhibited a significant correlation between GPR158 protein levels and immobility in the TST (Pearson $R^2$ = 0.4605, p=0.0014). (**B**) Mice injected acutely with corticosteroid (Acute Cort., 20 mg/kg) show no difference in GPR158 levels compared to vehicle (Veh) treatment (n = 5, Student's *t* test; ns, not significant). (**C**) Western blot quantification of GPR158 in primary cortical neurons cultured for 21 days in vitro and treated with 0.5 μM dexamethasone (Dex) from DIV 14 to DIV21 (n = 3, Student's *t* test, *p<0.05). (**D**) GPR158 protein levels in the mPFC of mice subjected to physical restraint stress and treated with RU-486. To induce chronic stress mice underwent a 17 day PRS period. On the last three days of the stress paradigm, mice were injected with RU-486 (10 mg/kg) or vehicle prior to being subjected to PRS. The mPFC was isolated 30 min following last episode of PRS (day 17). Stressed mice treated with RU-486 have a decrease in GPR158 protein levels compared to saline treatment (n = 4–5 mice/group, Two-way ANOVA with Tukey post hoc test, **p<0.01, ***p<0.001). Data shown as means ± SEM (Veh, vehicle; Ctrl, control; PRS, physical restraint stress; Cort., corticosterone).
DOI: https://doi.org/10.7554/eLife.33273.006

aspects related to mood and anxiety we additionally calculated an aggregate 'emotionality score' by meta-analysis of all behavioral data by z-scoring methodology (*Guilloux et al., 2011*) (*Figure 4H*). This analysis further revealed that mice injected with GPR158-Ad had a higher emotionality score compared to EGFP-Ad mice indicating that elevating GPR158 levels in the mPFC is sufficient to induce depressive-like behaviors.

To assess the clinical relevance of GPR158 expression modulation, we analyzed GPR158 protein levels in the postmortem dorsolateral PFC (dlPFC) of subjects diagnosed with MDD. Western blot analysis revealed a significant elevation in GPR158 protein levels in the dlPFC of MDD patients compared to match controls (*Figure 4J*; *Figure 4—figure supplement 1*), suggesting that this receptor may be involved in the neuropathology of depression.

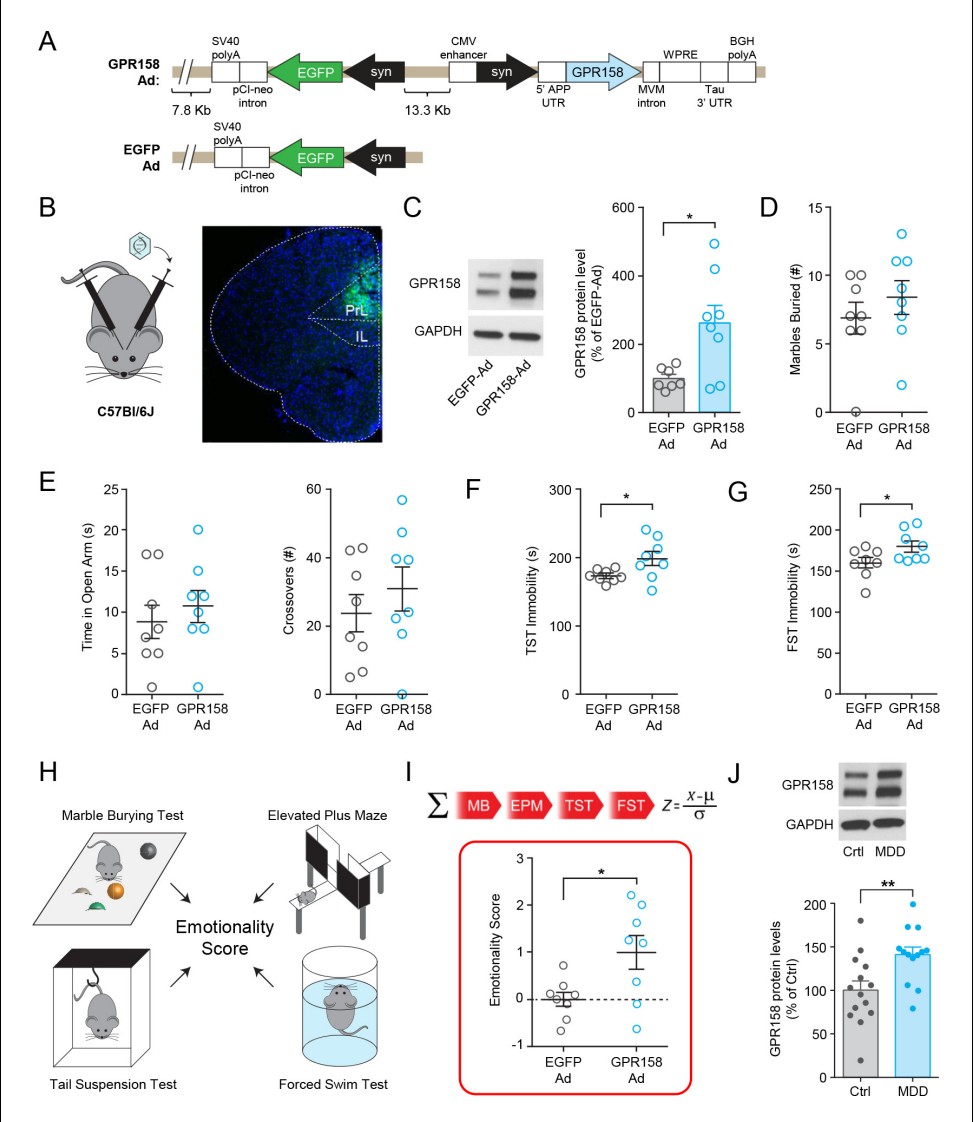

**Figure 4.** GPR158 overexpression in mouse mPFC and in dlPFC of MDD patient. (**A**) Scheme of Punisher-Adenovirus expressing GPR158 (GPR158-Ad) and EGFP (EGFP-Ad) injected bilaterally into the mPFC of C57Bl/6J mice. (**B**) Representative viral expression in the mPFC (right panel). (**C**) Representative western blots and quantification of GPR158 protein levels of mice injected with GPR158-Ad and EGFP-Ad. Injected mice underwent marble burying test (**D**), Elevated Plus Maze (**E**), Tail Suspension Test (**F**), and Forced Swim Test (**G**) (n = 8 mice/group). (**H**) Emotionality scores were integrated from four behavioral tests (marble burying, elevated plus maze, tail suspension test and force swim test) and normalized to mice injected with EGFP-Ad. (**I**) Mice injected with GPR158-Ad showed a higher emotionality score reflective of a depressive-like phenotype. (**J**) Representative western blot and quantification of GPR158 protein levels in the dlPFC show an increase in subjects with MDD compared to healthy unaffected controls (Ctrl) (Ctrl n = 14; MDD n = 13). The different groups were matched as closely as possible for sex, age, and postmortem interval (see Materials and methods for further information). Data are mean ± SEM (Student's *t* test, *p<0.05, **p<0.01).

DOI: https://doi.org/10.7554/eLife.33273.007

The following figure supplement is available for figure 4:

**Figure supplement 1.** GPR158 expression levels in patients affected by MDD.

DOI: https://doi.org/10.7554/eLife.33273.008

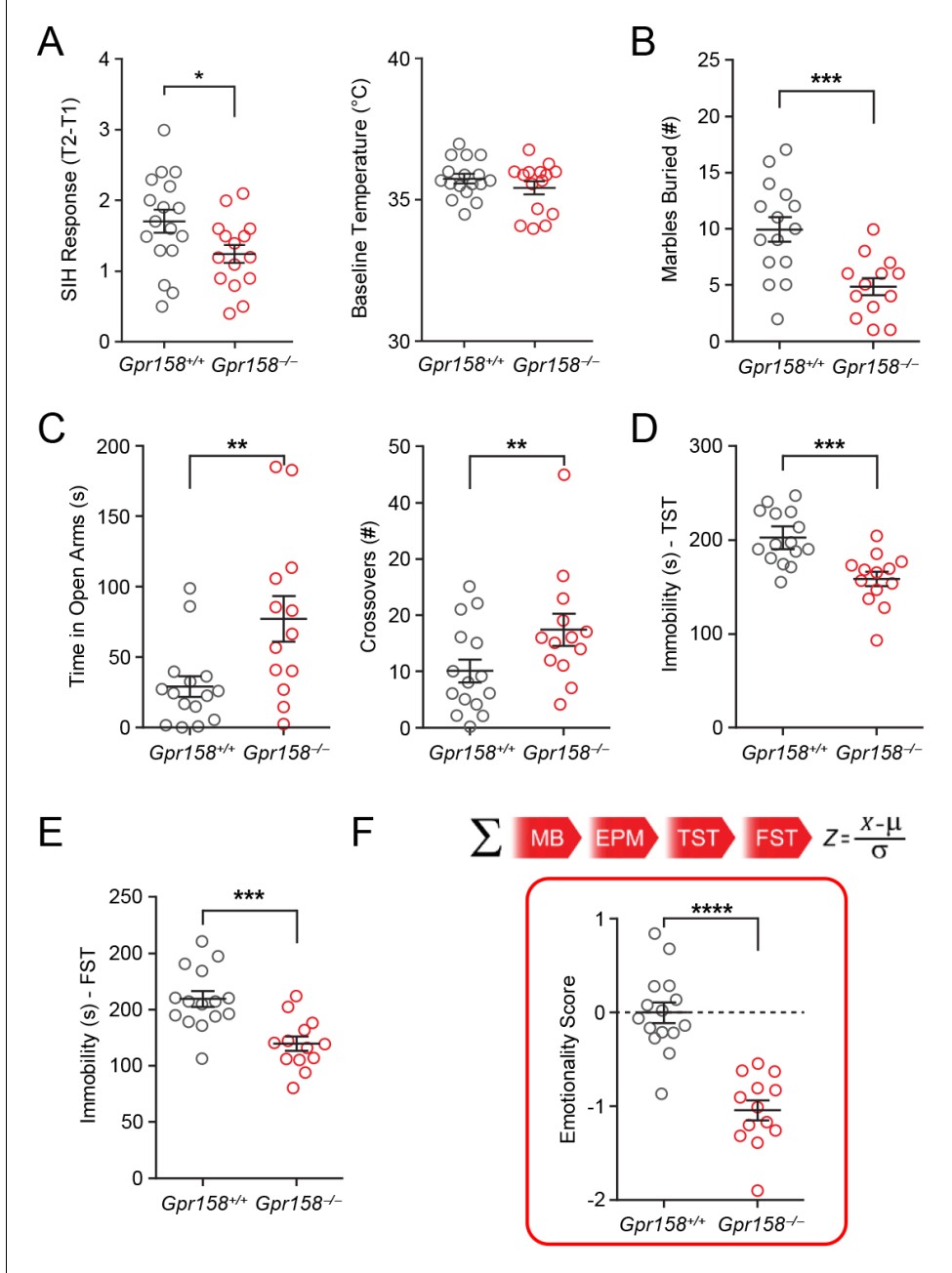

**Figure 5.** Ablation of GPR158 in mice results in an antidepressant-like phenotype and resiliency to chronic stress.
(**A**) *Gpr158*−/− mice display a lower SIH response (T2–T1) compared to *Gpr158*+/+ mice without affecting their
baseline temperature (right panel) (n = 15–17/genotype, Student's t test, *p<0.05). (**B**) *Gpr158*−/− mice buried
fewer marbles compared to *Gpr158*+/+ mice in the marble burying test. (**C**) *Gpr158*−/− mice spent significantly
more time in the open arm and entered more frequently in the open arm during the EPM test. *Gpr158*−/− mice
displayed decreased immobility time in the (**D**) TST and (**E**) FST. (**F**) Emotionality scores were integrated from four
behavioral tests (marble burying, elevated plus maze, tail suspension test and force swim test) and normalized to
*Gpr158*+/+ mice. (**G**) *Gpr158*−/− mice had a lower emotionality score reflective of an antidepressant-like phenotype
(right panel; Student's *t* test). Data shown as means ± SEM (*p<0.05, **p<0.01, ***p<0.001, ****p<0.0001; ns = not
significant).

DOI: https://doi.org/10.7554/eLife.33273.009

The following figure supplements are available for figure 5:

**Figure supplement 1.** Depressive- and anxiety-related behaviors are altered in both male and female *Gpr158*−/−
mice.

*Figure 5 continued on next page*

*Figure 5 continued*

DOI: https://doi.org/10.7554/eLife.33273.010

**Figure supplement 2.** Viral-mediated expression of GPR158 in mPFC of *Gpr158*$^{-/-}$ mice rescues the antidepressant-like phenotype.

DOI: https://doi.org/10.7554/eLife.33273.011

## Elimination of GPR158 in mice promotes an antidepressive-like phenotype

To further test the function of GPR158 in vivo, we evaluated the behavior of mice lacking GPR158 expression (*Gpr158*$^{-/-}$). *Gpr158*$^{-/-}$ animals showed attenuated response to stress-induced hyperthermia compared to their wild-type littermates, *Gpr158*$^{+/+}$ mice (*Figure 5A*), suggesting its involvement in stress-related behaviors. Male mice were evaluated in a panel of behavioral tests that assess various aspects of anxiety/depressive-like behaviors (*Figure 5B–E*). *Gpr158*$^{-/-}$ mice buried fewer marbles in the marble burying test (*Figure 5B*), spent more time and had more crossovers into the open arm in the elevated plus maze (*Figure 5C*), had reduced immobility in the tail suspension (*Figure 5D*) and forced swim tests (*Figure 5E*). Overall, the *Gpr158*$^{-/-}$ mice displayed a lower emotionality score compared to *Gpr158*$^{+/+}$ mice, corresponding to an anxiolytic and antidepressant-like phenotype (*Figure 5F*). These behavioral effects were specific as ablation of *Gpr158* did not affect overall behavioral reactions of mice including locomotor activity, habituation, motor coordination and learning (*Figure 5—figure supplement 1A–B*). Similarly, female *Gpr158*$^{-/-}$ mice also displayed a marked anxiolytic and antidepressant-like phenotype across behavioral tests demonstrating the behavioral effect due to the absence of GPR158 is independent of sex (*Figure 5—figure supplement 1C–G*). Furthermore, re-expression of GPR158 in the mPFC of adult *Gpr158*$^{-/-}$ mice by utilizing the GPR158-expressing viral vector (GPR158-Ad) restored the depressant-like phenotype as evidence by an increase in their emotionality score (*Figure 5—figure supplement 2A–F*).

To further explore the role of GPR158 in stress-induced depression we utilized the UCMS model. As expected, *Gpr158*$^{+/+}$ mice exposed to UCMS buried significantly more marbles in the marble burying test (*Figure 6A*), spent more time and had more crossovers into the open arm in the elevated plus maze (*Figure 6B*), exhibited increased immobility in the tail suspension (*Figure 6C*) and forced swim tests (*Figure 6D*). Together, *Gpr158*$^{+/+}$ mice that underwent UCMS displayed a higher emotionality score, consistent with the induction of a depressive-like state (*Figure 6E*). In contrast, *Gpr158*$^{-/-}$ mice were unaffected by UCMS, demonstrating a resiliency to chronic stress (*Figure 6A–E*). Furthermore, UCMS significantly reduced sucrose preference in *Gpr158*$^{+/+}$ mice but had no effect on *Gpr158*$^{-/-}$ mice demonstrating their resilience to stress-induced anhedonia (*Figure 6F*). Together, these results indicate the involvement of GPR158 in the control of stress-induced depression.

## GPR158 loss induces synaptic adaptations

To explore the mechanisms underlying the antidepressant-like effect associated with the elimination of GPR158 we first focused on examining changes in neuronal morphology. A morphological analysis of the glutamatergic neurons in the layers 2/3 of the prelimbic PFC revealed a 25% increase in spine density in *Gpr158*$^{-/-}$ mice compared to *Gpr158*$^{+/+}$ littermates (*Figure 7A*). Consistent with the observed increase in spine density, patch-clamp electrophysiological recordings revealed an increase in frequency of spontaneous excitatory post-synaptic currents (sEPSC) in the same neuronal population (*Figure 7B and C*). There were no significant changes in sEPSC amplitude (*Figure 7D*) and paired-pulse ratio (PPR) (*Figure 7—figure supplement 1A*) suggesting the postsynaptic nature of this effect (*Wu et al., 2007*).

We next determined the contribution of the AMPA and NMDA receptors to the generation of the postsynaptic response in layer 2/3 glutamatergic neurons because of their significant contributions to structural spine dynamics (*Ultanir et al., 2007*) and antidepressant effects (*Autry et al., 2011*). We found that the ratio of AMPAR to NMDAR components was significantly elevated in *Gpr158*$^{-/-}$ indicating that loss of GPR158 leads to an increase in the synaptic strength (*Figure 7E*).

To identify which receptor accounts for this effect, we performed glutamate uncaging experiments at single spines while measuring the currents mediated by each of the receptors individually

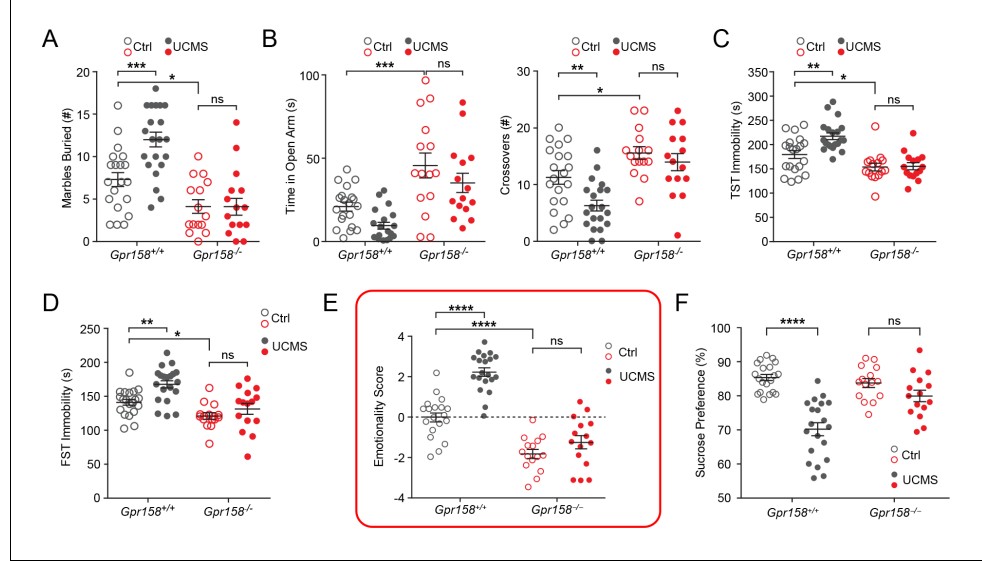

**Figure 6.** *Gpr158*$^{-/-}$ mice are resilient to unpredictable chronic mild stress (UCMS). Individual behavioral measures were integrated to calculate the emotionality score for each animal. Behavioral analysis of *Gpr158*$^{+/+}$ and *Gpr158*$^{-/-}$ mice subjected to UCMS in the (**A**) marble burying test, (**B**) elevated plus maze (EPM), (**C**) tail suspension test (TST) and (**D**) forced swim test (FST). (**E**) Emotionality score and (**F**) sucrose preference test for *Gpr158*$^{+/+}$ and *Gpr158*$^{-/-}$ mice subjected to UCMS (n = 15–20 mice/group, Two-way ANOVA with Bonferroni post hoc test). Data shown as means ± SEM (n = 15–20 mice/group; Two-way ANOVA with Bonferroni post hoc; *p<0.05, **p<0.01, ***p<0.001, ****p<0.0001; ns, not significant).
DOI: https://doi.org/10.7554/eLife.33273.012

(*Figure 7F–G*). These experiments revealed significantly elevated amplitude of the AMPAR responses in *Gpr158*$^{-/-}$ neurons (*Figure 7H*) with no significant changes in NMDAR function (*Figure 7I*). The effect persisted regardless of the size of the dendritic spine being analyzed (*Figure 7—figure supplement 1B–C*). Accordingly, treatment with the AMPAR blocker NBQX reverted the antidepressant-like phenotype in *Gpr158*$^{-/-}$ mice but did not affect *Gpr158*$^{+/+}$ mice in the FST (*Figure 7J*). Notably, we also found an increase in the levels of GluA2 and in the phosphorylation of GluA1 at Ser845 in mPFC of *Gpr158*$^{-/-}$ mice (*Figure 7K*), but no changes in total or phosphorylated NMDAR subunits were found (NR1, NR2B; *Figure 7—figure supplement 1D*). Considering that the phosphorylation at the Ser845 of GluA1 is mediated by PKA (*Roche et al., 1996*), we hypothesized that GPR158 may exert its effects through cAMP. Consistent with this expectation, we found significantly higher levels of cAMP in the mPFC of *Gpr158*$^{-/-}$ mice (*Figure 7L*).

To investigate the molecular basis behind the cellular adaptations and antidepressant-like phenotype we screened for changes in several signaling proteins induced by GPR158 loss in the mPFC (*Figure 8A–B*). Remarkably, we found a substantial increase in the levels of brain derived neurotrophic factor (BDNF) (*Figure 8B*), a hallmark feature thought to mediate the effects of antidepressants and their effects on spine morphology (*Duman and Monteggia, 2006*) (*Berton and Nestler, 2006*). This result was replicated on new cohort of mice (*Figure 8C*). Addressing the mechanisms involved we found no change in *Bdnf* mRNA levels but we observed a decrease in the phosphorylation level of the eukaryotic elongation factor 2 (eEF2), suggesting that the increase BDNF levels in *Gpr158*$^{-/-}$ mPFC may be due to local synthesis (*Figure 8D–E*). Finally, we found no change in monoamine neurotransmitter levels, or turnover, in the mPFC region of *Gpr158*$^{-/-}$ mice (*Figure 8F–G*), indicating that the behavioral effects observed in *Gpr158*$^{-/-}$ mice are due to alterations in receptor signaling mechanisms rather than effects on neurotransmitter activity. Overall, these results indicate that behavioral changes induced by GPR158 loss are paralleled by synaptic adaptations that potentiate structural and functional state of synapses.

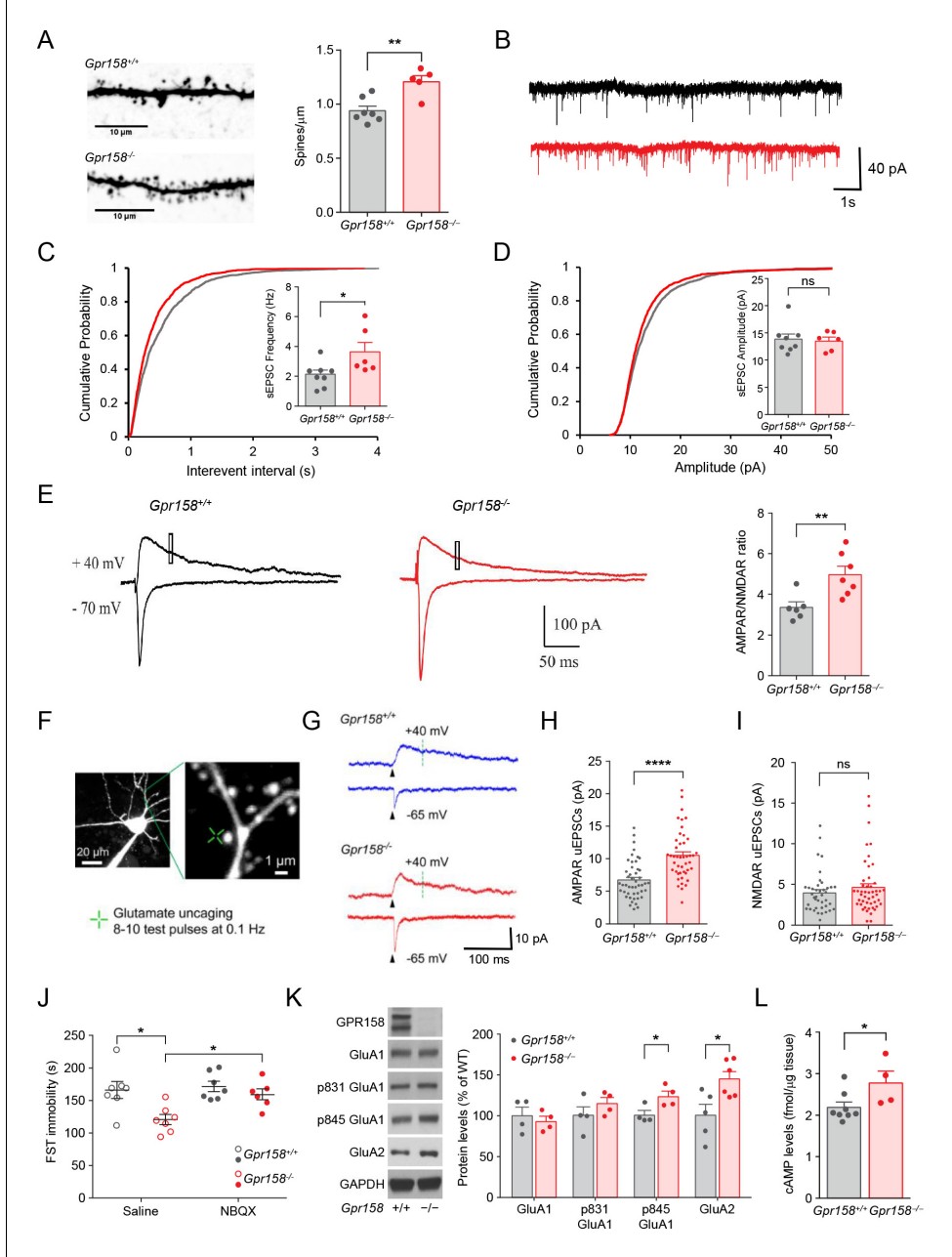

**Figure 7.** Synaptic adaptations in layer 2/3 neurons of mPFC induced by GPR158 loss. (A) Representative images and summarized data showing that the spine density was increased in $Gpr158^{-/-}$ neurons (n = 7–5). (B) Representative traces and summarized data showing that the frequency (C) (n = 8–6 mice/genotype; Student's *t* test), but not the amplitude (D) (n = 8–6 mice/genotype; Student's *t* test), of sEPSC was increased in $Gpr158^{-/-}$ neurons. (E) Representative traces and summarized data showing that AMPAR/NMDAR current ratio was increased in $Gpr158^{-/-}$ neurons (n = 6–7 mice/genotype). The AMPAR component was measured as the maximal response while neurons were held at −70 mV. The NMDAR component was measured as the average current between 50–55 ms (square) following the stimulation while the neurons were held at +40 mV. (F) Representative image of a spine and dendrite showing location of glutamate uncaging. (G) AMPAR- and NMDAR-mediated uEPSCs at −65 and +40 mV, respectively. The green dotted line (70 ms after uncaging pulse) indicates the time at which the NMDAR-uEPSCs were measured. Summary of (H) AMPAR uEPSC (n = 47–48 neurons/genotype) and (I) NMDAR uEPSC (n = 40–47 neurons/genotype) amplitude are shown for each genotype. (J) FST for $Gpr158^{-/-}$ and $Gpr158^{+/+}$ mice injected with NBQX (K) Representative western blots and quantification of total and phosphorylated levels of AMPAR subunits, GluA1 and GluA2 (n = 4–6). (L) Basal levels of cAMP in the mPFC of

*Figure 7 continued on next page*

*Figure 7 continued*

*Gpr158$^{+/+}$* and *Gpr158$^{-/-}$* mice (n = 4–8 mice/genotype). Data are mean ± SEM (Student's *t* test, *p<0.05, **p<0.01, ****p<0.0001, ns = not significant).
DOI: https://doi.org/10.7554/eLife.33273.013

The following figure supplement is available for figure 7:

**Figure supplement 1.** AMPA receptors are involved in the synaptic adaptations of *Gpr158$^{-/-}$* mice.

DOI: https://doi.org/10.7554/eLife.33273.014

## Discussion

In this study, we identify the orphan receptor GPR158 as a critical regulator of behavioral responses to stress. Since stress is a significant risk factor for MDD, understanding the cellular and molecular factors underlying behavioral vulnerability or resilience to stress-induced behavioral responses is poised to shed light on mechanisms of MDD initiation and progression. Several lines of evidence obtained in this study suggest that orphan receptor GPR158 may be one of the key molecular factors determining an individual's susceptibility or resiliency to stress-induced depression. The vast majority of evidence was obtained studying responses of mice to stress-induced depression, validated to model salient features of MDD including anhedonia, anxiety-like behaviors and physiological changes (*Covington et al., 2010*; *Elizalde et al., 2008*). We found that suppression of GPR158 produced an antidepressant-like phenotype and promoted a resiliency to stress. In contrast, induction of GPR158 induced a depressive-like effect. In addition to these causative effects, we found a strong correlation between levels of GPR158 and the emotional state of the animal in a way serving as a 'biomarker' of maladaptive responses. Notably, this relationship was further observed in humans where subjects diagnosed with MDD were found to also have elevated GPR158 levels as compared to unaffected individuals, although it is unknown if stress was a contributing factor. Mechanistically, stress induces changes in GPR158 expression via glucocorticoids. It is well established that prolonged glucocorticoid release precipitated by chronic stress leads to changes in transcription of multiple genes through direct engagement of nuclear receptor complexes and that this regulation plays a key role in the long-term structural and functional changes associated with mood disorders (*Papadopoulou et al., 2015*; *Schmidt et al., 2013*; *Yang et al., 2004*). Accordingly, we found that GPR158 upregulation requires sustained corticosteroid elevation induced by chronic but not acute stress exposure. Such differential sensitivity is thought to be underpinned by several mechanisms including epigenetic control and feedback at the glucocorticoid receptors (*Hunter et al., 2009*). Together, these findings make GPR158 bona fide glucocorticoid responsive gene in the brain.

Anatomically, we pinpoint that GPR158 exerts its effects on depressive-like behaviors in mPFC. Although the involvement of other regions in mediating the actions of GPR158 on depression-like and stress related behaviors cannot be ruled out, its effects in PFC appear to be sufficient for driving behavioral changes supported by both gain and loss of function experiments. Given GPR158 broad expression pattern throughout the brain, this orphan receptor may be involved in regulating other physiological functions. As such, GPR158 has been shown to play a role in hippocampal-memory function, a process mediated by osteocalcin (*Khrimian et al., 2017*). At this time, it is unknown if osteocalcin is involved in mediating the effects of GPR158 in stress-induced depression and further investigation is warranted. It is unclear whether osteocalcin is the only endogenous ligand of GPR158 across the entire brain or whether different ligands may be involved on modulation GPR158 activity in a region-specific manner. The latter possibility appears to be possible given that signaling changes that we observe in the PFC neurons are consistent with GPR158 serving as an inhibitory GPCR as opposed to Gq-coupled excitatory action in the hippocampus. In any event, it is clear that changes in the levels of GPR158 in the mPFC contribute to modulation of emotionality responses. Intriguingly, the mPFC is a key region targeted by stress hormones, including glucocorticoids and that stress-induced changes in mPFC are firmly linked to neuroendocrine and behavioral responses to stress. Manipulation of glutamatergic neurons in the mPFC have been shown to control behavioral responses to stressful events (*Kumar et al., 2013*; *Schmidt et al., 2012*; *Shrestha et al., 2015*), while optogenetic stimulation induced an antidepressant-like response (*Covington et al., 2010*; *Fuchikami et al., 2015*). Studies in animal models indicate that loss of mPFC control drives anxiety- and depressive-like phenotypes (*Amat et al., 2005*; *Shrestha et al., 2015*), while treatments that

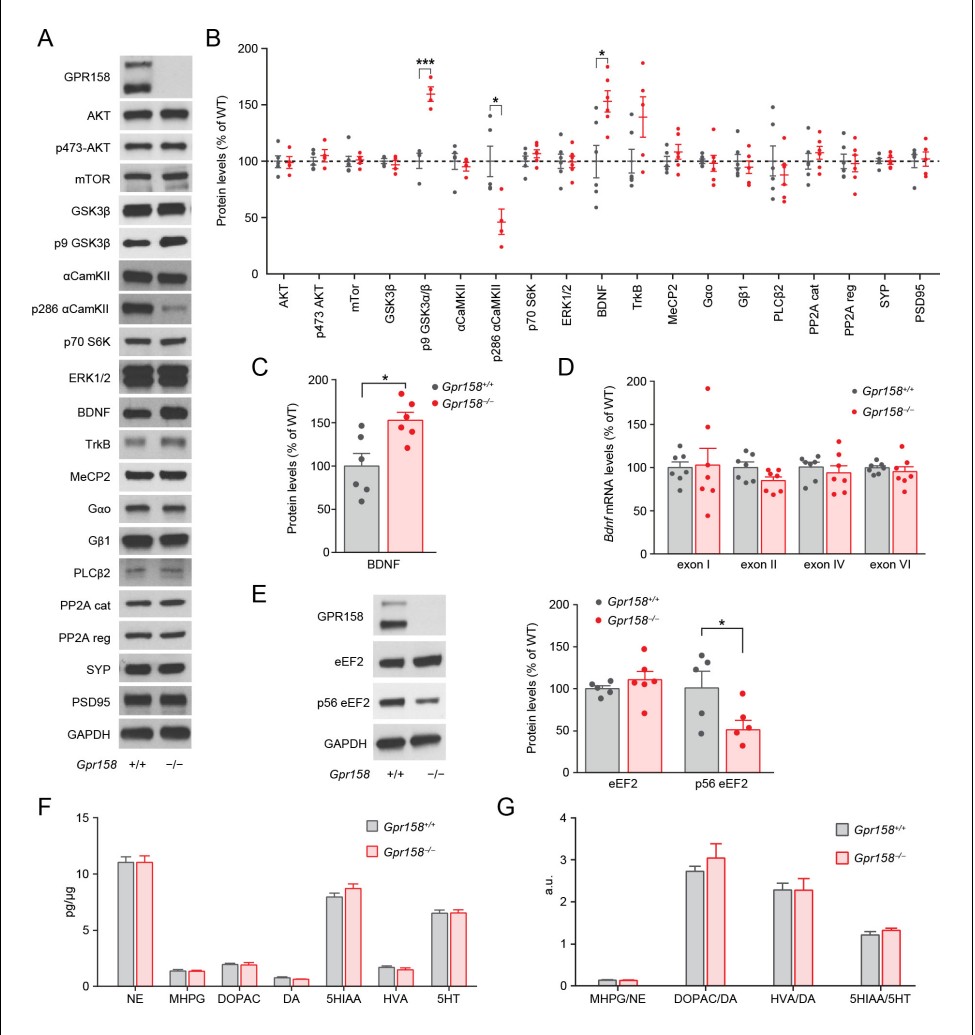

**Figure 8.** GPR158 modulates BDNF protein levels in the mPFC. Representative western blots (**A**) and quantification (**B**) of the protein levels and phosphorylation state of the indicated proteins in the mPFC of *Gpr158⁺/⁺* and *Gpr158⁻/⁻* mice (n = 4–6). (**C**) Western blot quantification of BDNF levels in the mPFC of *Gpr158⁺/⁺* and *Gpr158⁻/⁻* mice (n = 6). (**D**) Comparison of mRNA levels of 4 BDNF splicing isoforms in the mPFC of *Gpr158⁺/⁺* and *Gpr158⁻/⁻* do not show any significant difference. GPR158 levels were normalized against the ribosomal subunit 18S RNA (n = 7). (**E**) Western blot and quantification of total and phosphorylated levels of eEF2 (n = 5–6). (**F**) Quantitative analysis of the levels of norepinephrine (NE), 3-methoxy-4-hydroxyphenylglycol (MHPG), 3,4-dihydroxyphenylacetic acid (DOPAC), dopamine (DA), 5-hydroxyindoleacetic acid (5HIAA), homovanillic acid (HVA) and serotonin (5HT) in the mPFC of *Gpr158⁺/⁺* and *Gpr158⁻/⁻* (**G**) Neurotransmitter turnover calculated as metabolite/neurotransmitter ratio. Data are mean ± SEM (Student's *t* test, *p<0.05, ***p<0.001).
DOI: https://doi.org/10.7554/eLife.33273.015

act within the mPFC promote a therapeutic effect (*Kumar et al., 2013*). Furthermore, we provide evidence that within PFC, GPR158 modulates properties of the glutamatergic neurons in layer 2/3. Intriguingly, recent studies provide strong support for a role of layer 2/3 in the PFC in stress and depression. Stress-induced depression was documented to reduce spine densities and branch points in layer 2/3 of the mPFC (*Radley et al., 2008*; *Radley et al., 2004*) whereas activation of their sub-population induced rapid and distinct changes in depression-like behaviors (*Warden et al., 2012*). Given that these neurons innervate various limbic and brainstem structures, and serve as the main projections for intracortical communication (*Gabbott et al., 2005*) it seems possible that modulation of stress and depression-like behaviors by GPR158 to some extent involves its effects on layer 2/3 neurons.

At the cellular level, we determined that GPR158 exerts its effects by profoundly influencing the synaptic strength. These changes are evident at both the structural and functional level as reflected by increase in spine density and enhanced AMPAR/NMDAR ratio, respectively. We found that these changes are largely attributed to an increase in postsynaptic AMPAR function due to an increase in both GluA2 expression and GluA1 phosphorylation at Ser845 which increases its content on the surface (*Esteban et al., 2003*) and synaptic strength (*Maya Vetencourt et al., 2008*). Accordingly, our findings establish GPR158 as a critical negative regulator of synaptic strength, which is likely mediating its cellular and behavioral effects. In addition, to the direct functional effects on processing synaptic inputs, increase in AMPAR signaling has been also shown to increase the expression of key neurotrophic factor BDNF via local protein synthesis (*Jakawich et al., 2010*; *Zanos et al., 2016*). Although the actions of BDNF are complex and not fully understood, it is thought to further exert anti-depressant effects through structural alterations, transcriptional and epigenetic changes (*Björkholm and Monteggia, 2016*). Accordingly, our findings establish GPR158 as a critical negative regulator of both synaptic strength and BDNF production, which are likely key effector systems mediating its cellular and behavioral effects. Several lines of evidence support the role of synaptic adaptations and AMPAR function in depression, stress and the development of MDD. For example, reduction in AMPA receptor-dependent synaptic responses in the PFC has been observed in models of stress-induced depression (*Yuen et al., 2012*). Conversely, ketamine and its metabolites exert rapid and powerful antidepressant effects in animal models of depression (*Autry et al., 2011*; *Parise et al., 2013*) and in humans (*Zarate et al., 2006*) by increasing synaptic strength defined as upregulation in AMPAR/NMDAR ratio by either blocking NMDAR function (*Miller et al., 2014*) or augmenting AMPAR activity (*Zanos et al., 2016*), respectively. In accordance with our GPR158 results, the antidepressant effects of ketamine or traditional antidepressant medication is blocked by the inhibition of AMPAR (*Gould et al., 2008*; *Zhou et al., 2014*). Finally, chronic treatments with typical antidepressants have been shown to affect AMPA receptor levels in the PFC (*Barbon et al., 2011*). The increased cAMP levels observed in the mPFC of *Gpr158*$^{-/-}$ mice suggest that cAMP-dependent PKA activation may be responsible for the increased phosphorylation of GluA1. Thus, GPR158 may function at the critical nexus of stress, depression, and synaptic plasticity linking these processes at the molecular level.

In summary, our data demonstrates that GPR158 plays a prominent role in the regulation of stress-induced depression via adaptations in synaptic plasticity. These findings implicate GPR158 signaling in the pathophysiology of depression and identify a new therapeutic target to block or reverse the effects of chronic stress and depressive behaviors.

# Materials and methods

## Key resources table

| Reagent type (species) or resource | Designation | Source or reference | Identifiers | Additional information |
|---|---|---|---|---|
| Antibody | GAPDH | Millipore | Cat# AB2302 | (1:30000) |
| Antibody | GluA1 | Abcam | Cat# ab76321 | (1:1000) |
| Antibody | p831 GluA1 | Millipore | Cat# 04–823 | (1:1000) |
| Antibody | p845 GluA1 | Cell Signaling | Cat# 8084 | (1:1000) |
| Antibody | GluA2 | Millipore | Cat# MAB N71 | (1:1000) |
| Antibody | NR1 | Zymed | Cat# 32–500 | (1:1000) |
| Antibody | p897 NR1 | Millipore | Cat# 06–641 | (1:1000) |
| Antibody | NR2B | Millipore | Cat# AB1557P | (1:1000) |
| Antibody | Rabbit anti-GPR158CT | PMID:25792749 | N/A | (1:1000) |
| Antibody | Rabbit anti-Gβ5 | PMID:15632198 | N/A | (1:1000) |
| Antibody | Rabbit anti-Gβ1 | gift from Dr. Willardson PMID:15485848 | N/A | 1:6000 |
| Antibody | GPR34 | ThermoFisher Scientific | Cat# PA5-45717 | (1:1000) |

*Continued on next page*

*Continued*

| Reagent type (species) or resource | Designation | Source or reference | Identifiers | Additional information |
|---|---|---|---|---|
| Antibody | GPR162 | Aviva Systems Biology | Cat# ARP68400 | (1:1000) |
| Antibody | GPRC5B | Aviva Systems Biology | Cat# ARP59799 | (1:1000) |
| Antibody | AKT | Cell Signaling | Cat# 9272 | (1:1000) |
| Antibody | p473 AKT | Cell Signaling | Cat# 4060 | (1:1000) |
| Antibody | mTOR | Cell Signaling | Cat# 2983 | (1:1000) |
| Antibody | S6 Kinase (P70) | Cell Signaling | Cat# 9202 | (1:1000) |
| Antibody | ERK1/2 | Cell Signaling | Cat# 9102 | (1:1000) |
| Antibody | GSK-3β | Cell Signaling | Cat# 9315 | (1:1000) |
| Antibody | p9 GSK3α/β | Cell Signaling | Cat# 9331 | (1:1000) |
| Antibody | BDNF | Abcam | Cat# ab108319 | (1:1000) |
| Antibody | TrkB | Cell Signaling | Cat# 4603 | (1:1000) |
| Antibody | MeCP2 | Millipore | Cat# 07–013 | (1:1000) |
| Antibody | Gαo | Cell Signaling | Cat# 3975 | (1:1000) |
| Antibody | PLCβ2 | Santa Cruz | Cat# sc206 | (1:200) |
| Antibody | PP2Acat | Assay Biotech | Cat# B0555 | (1:1000) |
| Antibody | PP2Areg | Cell Signaling | Cat# 2041 | (1:1000) |
| Antibody | SYP | Assay Biotech | Cat# C0333 | (1:1000) |
| Antibody | PSD95 | Cell Signaling | Cat# 3450 | (1:1000) |
| Antibody | eEF2 | Cell Signaling | Cat# 2332 | (1:1000) |
| Antibody | p56 eEF2 | Cell Signaling | Cat# 2331 | (1:1000) |
| Antibody | NeuN | Abcam | Cat# 104225 | (1:1000) |
| Antibody | GAD67 | Millipore | Cat# MAB5406 | (1:1000) |
| Genetic reagent (adenovirus) | HdAd-23E4-Pun-GPR158-syn-EGFP | This Paper | N/A | |
| Genetic reagent (adenovirus) | HdAd-23E4-Pun-Syn-EGFP | This Paper | N/A | |
| Biological sample (brain tissue) | Adult human brain tissue (healthy and MDD samples) | NIH NeuroBioBank (University of Pittsburg, University of Miami Brain Endowment Bank and the Human Brain and Spinal Fluid Resource Center) | 13138, 13054, 13082, 5289, 002, 006, 5293, AN04642, AN04917, AN07465, AN14368, AN15392, AN01077, AN05364, 13006, 13011, 13022, 13047, 13075, 13086, 13145, 13057, 13051, 13181, 535, 556, 711 | |
| Chemical compound, drug | Corticosterone | Sigma-Aldridge | Cat# C2505 | |
| Chemical compound, drug | RU-486 | Sigma-Aldridge | Cat# M8046 | |
| Chemical compound, drug | Dexamethasone | Tocris | Cat# D4902 | |
| Chemical compound, drug | NBQX | Tocris | Cat# 10441R | |
| commercial assay or kit | QuantiGene ViewRNA ISH Tissue 2-Plex Assay | Affymetrix | Cat# QVT0012 | |
| commercial assay or kit | *Gpr158* NM_001004761; ViewRNA TYPE 1 Probe | Affymetrix | Cat# VB1-11518 | |
| commercial assay or kit | *Slc17a7* NM_182993; ViewRNA TYPE 6 Probe | Affymetrix | Cat# VB6-14162 | |
| commercial assay or kit | *Gad1* NM_008077; ViewRNA TYPE 6 Probe | Affymetrix | Cat# VB6-12632 | |

*Continued*

| Reagent type (species) or resource | Designation | Source or reference | Identifiers | Additional information |
|---|---|---|---|---|
| commercial assay or kit | *Pvalb* NM_013645; ViewRNA TYPE 6 Probe | Affymetrix | Cat# VB6-13220 | |
| commercial assay or kit | *Sst* NM_009215; ViewRNA TYPE 6 Probe | Affymetrix | *Cat# VB6-13754* | |
| commercial assay or kit | *DapB* NC_000913; ViewRNA TYPE 1 Probe | Affymetrix | Cat# VF1-10272 | |
| commercial assay or kit | Direct cAMP ELISA kit | ENZO Life Sciences | Cat# ADI-900–066 | |
| strain, strain background (mus musculus) | GPR158 KO | PMID:25792749 | N/A | |
| software, algorithm | ImageJ | Wayne Rasband, NIH, USA | SCR_003070 | |
| software, algorithm | Zen2.1 SP1 (Black) | Carl Zeiss | SCR_013672 | |
| software, algorithm | Prism6 | GraphPad Software | SCR_002798 | |
| software, algorithm | Neurolucida | MBF Bioscience | | |

## Experimental models and subject details

All studies were carried out in strict accordance with the recommendations in the Guide for the Care and Use of Laboratory Animals of the National Institutes of Health. All procedures were approved by the Institutional Animal Care and Use Committee (IACUC) protocol (#16–032) at The Scripps Research Institute. The $Gpr158^{-/-}$ mice were purchased from KOMP ($Gpr158^{tm1(KOMP)Vlcg}$) and maintained on a C57/Bl6 background. They were breed as heterozygous breeding pairs to generate $Gpr158^{-/-}$ and $Gpr158^{+/+}$ littermates. C57Bl/6J mice were used in biochemical and behavioral experiments using stress-induced paradigms described below. We relied exclusively on littermates for all the comparisons. Mice were housed in groups (unless otherwise stated) on a 12 hr light-dark cycle with food and water available *ad libitum*. Both male and female $Gpr158^{-/-}$ mice were used to assess emotionality. Male mice were used in all other behavioral assays, while both male and female were used for biochemistry analysis, and were between 2–5 months of age.

## Antibodies and western blots

For western blot analysis, brains were quickly removed from euthanized mice and medial prefrontal cortex was excised with a 3 mm puncher. Tissues were lysed in ice-cold lysis buffer (300 mM NaCl, 50 mM Tris-HCl pH 7.4, 1% Triton X-100, complete protease inhibitor cocktail (Roche Applied Science, Penzberg, Germany) and phosphatase inhibitor mix (Sigma-Aldrich, St. Louis, MO) by sonication, incubated on a rocker for 30 min at 4°C and cleared by centrifugation at 14,000 rpm for 15 min. The supernatant was saved and the protein concentration was obtained using Pierce 660 nm Protein Assay (Thermo Fisher, Waltham, MA). Samples were diluted in 4 × SDS sample buffer and analyzed by SDS-PAGE. Researcher was blinded to the genotype/treatment history of the samples.

The generation of rabbit antibodies against Gpr158CT was described earlier (*Orlandi et al., 2015*). Rabbit anti-Gβ1 was a kind gift from Dr. Barry Willardson (Brigham Young University, Provo, Utah). The following antibodies used for western blot were purchased: GPR34 (Thermo; PA5-45717); GPR162 (Aviva; ARP68400); GPRC5b (Aviva; ARP59799); GAPDH (Millipore; MAB374); AKT (Cell Signaling; 9272); p473 AKT (Cell Signaling; 4060); mTOR (Cell Signaling; 2983); S6 Kinase (P70) (Cell Signaling; 9202); ERK1/2 (Cell Signaling; 9102); GSK-3β (Cell Signaling; 9315); p9 GSK3α/β (Cell Signaling; 9331); BDNF (Abcam; ab108319); TrkB (Cell Signaling; 4603); MeCP2 (Millipore; 07–013); Gαo (Cell Signaling; 3975); PLCβ2 (Santa Cruz; sc206); PP2Acat (Assay Biotech; B0555); PP2Areg (Cell Signaling; 2041); GluA1 (Abcam; ab76321); p831 GluA1 (Millipore; 04–823); p845 GluA1 (Cell Signaling; 8084); GluA2 (Millipore; MAB N71); NR1 (Zymed; 32–500); p897 NR1 (Millipore; 06–641); NR2B (Millipore; AB1557P); SYP (Assay Biotech; C0333); PSD95 (Cell Signaling; 3450); eEF2 (Cell Signaling; 2332); p56 eEF2 (Cell Signaling; 2331). The following antibodies used for immunohistochemistry were purchased: NeuN (Abcam; 104225); GAD67 (Millipore; MAB5406).

## Primary cortical cultures

Mouse cerebral cortices were extracted from neonatal P0-1 pups and placed into an ice-cold HBSS/FBS solution: Hank's Balanced Salt Solution (Sigma, St. Louis, MO), 4.2 mM NaHCO$_3$, 1 mM HEPES, and 20% FBS. The tissue was washed three times with HBSS and then digested at 37°C/15 min with Papain (Worthington Biochemical, Lakewood) in a solution that contained 137 mM NaCl, 5 mM KCl, 7 mM Na$_2$HPO$_4$, and 25 mM HEPES (pH 7.2). The tissue was washed 3 times with 20% FBS, three times with HBSS, and three times with growth medium containing Neurobasal A, B27, Glutamax and Penicillin/Streptomycin (Invitrogen, Carlsbad, CA). Tissue was then triturated in growth medium containing 50 units of DNAse I (Invitrogen, Carlsbad, CA). The neurons were pelleted by centrifugation (600 × g for 10 min) and plated at a density of 40,000 cells/cm$^2$ on poly-D-lysine-coated (Sigma-Aldrich, St. Louis, MO) 60 mm dishes. Neurons were incubated at 37°C/5% CO$_2$, and half of the medium was replaced with fresh medium once a week. 0.5 μM dexamethasone (Sigma-Aldrich, St. Louis, MO) or vehicle (EtOH) was applied after 14 days in culture for 7 days.

## In situ hybridization and immunohistochemistry

The mRNA expression of *Gpr158* was evaluated with ViewRNA$^{TM}$ 2-plex In Situ Hybridization Assay (Panomics, Santa Clara, CA) using the following probe sets: *Gpr158* (NM_001004761; Cat# VB1-11518), *vGlut1* (NM _182993; Cat# VB6-14162), *Gad1* (NM_008077; Cat# VB6-12632), *Pvalb* (NM_013645; Cat# VB6-13220), *Sst* (NM_009215; Cat# VB6-13754). A probe against the E. Coli gene *DapB* (NC_000913; Cat# VF1-10272) was used as specificity control as recommended by manufacturer. Briefly, mouse brains were embedded in OCT, flash frozen in liquid nitrogen, cut in 14 μm coronal sections and rapidly fixed in 4% paraformaldehyde for 10 min. Sections were then washed and incubated for 2 hr/RT in pre-hybridization mix (50% deionized formamide, 5X SSC, 5X Denhardt's solution, 250 μg/mL yeast tRNA, 500 μg/mL sonicated salmon sperm DNA), followed by overnight incubation at 40°C with Panomics hybridization solution containing TYPE one and TYPE 6 QuantiGene ViewRNA probe sets diluted 1:100. Sections were then processed according to manufacturer's instructions. To identify the soma of the cells, each section was counterstained with Neuro-Trace 435/455 Blue Fluorescent Nissl Stain (1:100, Molecular Probes, Eugene OR) and mounted using Fluoromont-G (Southern Biotech, Birmingham, AL). When immunohistochemistry was performed in combination with in situ hybridization, the same protocol was followed without Nissl staining. AlexaFluor-488 and AlexaFluor-546-conjugated secondary antibodies (1:1000) were purchased from Invitrogen. DAPI was used for nuclear labeling.

All the images were acquired at The Light Microscopy Facility, the Max Planck Florida Institute, using on a LSM 780 Zeiss confocal microscope. Image acquisition and processing were accomplished using ZEN 2011 software (Carl Zeiss, Oberkochen, Germany) with only minor manipulations of the images setting the fluorescence intensity in non-saturating conditions and maintaining similar parameters for each acquired image.

## RNA extraction and quantitative Real-Time PCR

Total RNAs from mouse medial prefrontal cortex were extracted using Trizol reagent according to the manual (Invitrogen, Carlsbad, CA). The RNA present in the aqueous phase was further purified using RNeasy kit (Qiagen, Hilden, Germany). Reverse transcription was carried out using qScript cDNA Supermix (Quantabio, Beverly, MA) according to manufacturer's instructions using 700 ng of total RNA. The reaction mixture was incubated at 25°C for 5 min, at 42°C for 30 min and then the enzyme was inactivated at 85°C for 5 min. Real-time PCR was performed with StepOne Real-time PCR system (Applied Biosystems, Foster City, CA) using SYBR Green (Roche Diagnostics, Indianapolis, IN). Relative expression levels were calculated by the comparative CT method and normalized to 18S ribosomal RNA. As reported (*Vanevski and Xu, 2015*), RT-PCR primers used were:

*Bdnf* Exon I Forward: ACTGAGTCTCCAGGACAGCAAAG; Exon II Forward: GTGGTGTAAGCCGCAAAGAA; Exon IV Forward: CAGAGCAGCTGCCTTGATGTT;

Exon VI: CAGAAGCGTGACAACAATGTGA; common *Bdnf* Reverse: CCTTCATGCAACCGAAGTATGA. 18S Forward: ACCGCAGCTAGGAATAATGGA;

18S Reverse: GCCTCAGTTCCGAAAACCA.

## RNA sequencing

The 100 bp reads were generated by the HISeq Analyzer 2000 at the Scripps DNA Sequencing Facility. The Genome Analyzer Pipeline Software (Casava v1.8.2) was used to perform the early data analysis of a sequencing run, which does the image analysis, base calling, and demultiplexing. For RNA-Seq TopHat was used for alignment to the Mus_musculus genome Build mm10. For the mRNA gene annotation, Partek software, version 6.6 (Partek Inc., St. Louis, MO, USA) was used. The number of aligned reads for the three brain samples analyzed was ~30 million reads. These data were filtered to exclude transcripts for which the maximum number of reads for all samples was <40 reads eliminating noise and calculated infinite fold-changes. Normalization was performed by the TMM (trimmed mean of M) method where the sample whose 75%-ile (of library-scale-scaled counts) is closest to the mean of 75%-iles as the reference. This adjustment was applied to the library size (total sample counts) to account for the compositional bias.

## Mass spectrometry analysis

The protein samples were subjected to concentrating SDS-PAGE at 120 V for 13 min. The unique gel band obtained was visualized by Coomassie staining (GelCode Blue Stain Reagent, Thermo Scientific, Waltham, MA) for 1 hr at room temperature with constant shacking, followed by water destaining overnight, and then cut into small gel pieces and submitted to in-gel trypsin digestion. Briefly, the gel pieces were treated with 10 mM DTT followed 50 mM iodoacetamide at pH 8.0 prior tryptic digestion (Promega, Madison, WI) overnight at 37°C. The resulting peptide pool was acidified and desalted onto Oasis C18 HLB extraction cartridges (Waters, Milford, MA) and dried down. The peptides were resuspended in TEAB prior to high pH reversed-phase chromatography. The 60 fractions obtained were combined in 15 final fractions using a concatenation strategy, prior desalting using OMIX C18 tips (Varian, Inc., Palo Alto, CA). After elution, the peptides were dried-down and analyzed by reverse phase-high performance liquid chromatography (RP-HPLC)-Q-OT-qIT analysis.

All digested samples were analyzed by liquid-chromatography-tandem MS (LC-MS/MS) using an EASY-nLC 1000 system coupled to an Orbitrap Fusion Tribrid Mass Spectrometer (Thermo Fisher Scientific, Waltham, MA). Peptides were on-line eluted on an analytical RP column (0.075 × 250 mm Acclaim PepMap RLSC nano Viper, Thermo Fisher Scientific), operating at 300 nL/min using the following gradient: 5–25% B for 40 min, 25–44% B for 20 min, 44–80% B in 10 s, 80% B for 5 min, 80–5% in 10 s, and 5% B for 20 min [solvent A: 0.1% formic acid (v/v); solvent B: 0.1% formic acid (v/v), 80% $CH_3CN$ (v/v) (Fisher Scientific, Pittsburgh, PA)]. The Orbitrap Fusion was operated in a data-dependent MS/MS mode using the 10 most intense precursors detected in a survey scan from 380 to 1,400 m/z performed at 120K resolution. Tandem MS was performed by HCD fragmentation with normalized collision energy (NCE) of 30.0%.

Protein identification was carried out using ProLuCID algorithm(*Xu et al., 2015*) (Integrated Proteomics Pipeline v.3, Integrated Proteomics Applications INC., San Diego, CA), allowing optional modifications (Met oxidation, Asn/Gln deamidation), two missed cleavages, and mass tolerance of 10 and 20 ppm for precursor and fragment ions, respectively. MS/MS raw files were searched against a homemade database containing the sequence of the complete Mouse proteome, human keratins and porcine trypsin. The raw files were also searched against a reverse sequences database. The false discovery rates (FDR) of peptide identifications were calculated to be 4.36 and 1.59 at the protein and peptide level respectively.

## Stress models

Male mice, age 2–3 months old were used in all stress related experiments, All animals were age and weight matched before commencement of any stress paradigm and were randomly assigned into the stress group or non-stress group.

Unpredictable chronic mild stress paradigm: For the unpredictable chronic mild stress (UCMS) paradigm, mice were housed in individual cages and allowed 1 week to habituate before beginning experimentation after which mice were subjected to 4 weeks of stressors with two stressors applied within a 24 hr period. Stressors were applied pseudo-randomly and included intermittent bell (10 db, 1/10 s), continuous white noise (4 hr), rat odor, cage tilt (45°, 4 hr), cage shaking (30 min), soiled bedding (8 hr), paired housing (with new partner, 2 hr), overnight illumination, removal of nesting material (12 hr), placement of novel object in home cage (3 hr) and confinement in a small cage (80

cm$^3$, 1 hr). Stressors continued to be applied during the behavioral testing phase. No stressors were applied on the day of testing (18 hr prior to any behavioral testing) to avoid effects of fatigue and acute stress. Non-stressed mice were left undisturbed in their home cages. Biochemical analysis was performed using the mPFC of mice isolated 1 hr following the final exposure to UCSM. The diagram below summarizes the time course outlining when the behavioral tests were performed with respect to chronic stress:

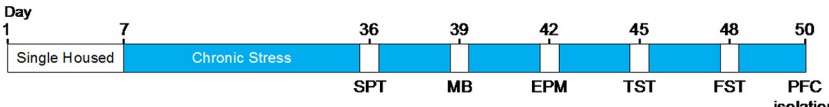

**Schema 1.** Timeline for behavioral evaluation in stress-induced depression models.
DOI: https://doi.org/10.7554/eLife.33273.016

Physical restraint stress paradigm: Physical restraint stress (PRS) was performed in plastic tube (30 mm diameter x 115 mm length) with holes for ventilation. Mice were restrained horizontally in tubes for 2 hr for 14 days. Non-stressed mice were left undisturbed in their home cages. During behavioral testing, mice were not subjected to PRS to avoid effects of fatigue and acute stress. Biochemical analysis was performed using the mPFC of mice isolated 1 hr following the final exposure to PRS.

Corticosterone paradigm: For the corticosterone study, pure corticosterone (Sigma-Aldridge, St. Louis, MO) was dissolved in 50% propylene glycol. Mice were administered corticosterone (20 mg/kg, s.c.) or vehicle for 14 days. Biochemical analysis was performed using the mPFC of mice following 1 hr post-injection time.

Stress-induced hypothermia: Mice undergoing stress-induced hypothermia (SIH) were housed singly overnight in the testing room with free access to food and water. Rectal temperature was measured twice in each mouse at $t = 0$ (T1) and $t = 15$ min (T2), where the difference between the temperatures (T2-T1) is considered the SIH score. Temperature was measured when a stable recording was observed, approximately 15 s following the insertion of the probe (ThermoWorks, American Fork, UT).

## Drug treatments

Mice were injected with saline on three consecutive days before the start of the study to acclimate the animals to the injection procedure. RU-486 (Sigma-Aldridge, St. Louis, MO) was injected at dose of 10 mg/kg for three consecutive days (*Yuen et al., 2012*) and tissue was isolated 30 min post-injection time. NBQX (Tocris Bristol, United Kingdom) was administered at a single dose of 10 mg/kg (i. p.) or appropriate vehicle and 30 min later were subjected to the behavioral testing.

## Recombinant helper dependent adenoviral production of GPR158 over-expression vector

GPR158 was first cloned into the high-level neuronal transgene expression cassette pUNISHER (*Montesinos et al., 2011*) for rapid and long term in vivo neuronal expression in the CNS. Briefly, this cassette includes the 365 bp human cytomegalovirus enhancer; 470 bp human *synapsin* promoter, the 107 bp 5' amyloid precursor protein untranslated region, the 92 bp minute virus of mice intron, 592 bp woodchuck posttranscriptional regulatory element, the 241 bp tau 3' UTR and the 224 bp bovine growth hormone polyadenylation signal. Subsequently, the expression cassette was cloned into the AscI site of pdelta 28E4, gift from Dr. Phil Ng (*Palmer and Ng, 2003*) using In Fusion, Clontech. This version of pdelta28E4 has been modified to contain 21 kbp of stuffer DNA and a separate neurospecific EGFP expression cassette that is driven by the hsyn promoter. The final HdAd plasmid allows for over-expression of GPR158 independently of EGFP as a dual expression recombinant Ad vector similar to the strategy used with 2$^{nd}$ generation rAd vectors which lead to no toxicity in neurons (*Chen et al., 2013*; *Montesinos et al., 2011*). Production of HdAd was carried out as previously described (*Palmer and Ng, 2008*). Briefly, HdAds were produced by first digesting the pHdAd with PmeI to linearize and expose the ends of the 5' and 3' ITRs. Transfection of the pHdAd was performed using 116 producer cells, a modified HEK293 line expressing high levels of the Cre recombinase enzyme, and a 4-kbp adenoviral genome fragment that encodes for the E1A/E1B gene, necessary for rAd to replicate (*Palmer and Ng, 2003*). The standard protocols for recombinant Helper-Dependent Adenovirus production were followed (*Palmer and Ng, 2008*) with slight

modifications (*Montesinos et al., 2016*). HdAd was serially amplified in five consecutive passages from 3 60 mm tissue culture dishes followed by one 15 cm dish and finally 30 15 cm dishes. Each successive passage was performed after cytopathic effect (CPE) occurred and cell lysates were subjected to three freeze/thaw cycles to lyse cells and thereby release the viral particles. HdAd was stored at −80°C in storage buffer (in mM): 10 HEPES, 250 sucrose, 1 MgCl$_2$ at pH 7.4 (*Croyle et al., 1998*). Viral particle/mL was calculated (*Palmer and Ng, 2008*) as follows:

Viral particles/ml = (A260) x (dilution factor) x ($1.1 \times 10^{12}$) x (36)/ (size of the vector in kb)

Viral Titer: HdAd 23E4 Pun GPR158 syn EGFP $7.11 \times 10^{12}$ vp/ml

## Behavioral measures

All behavioral experiments were carried out with the experimenter blind to genotype and/or treatment history.

### Marble Burying

Marble burying (MB) was conducted in a homecage-like environment ($27 \times 16.5 \times 12.5$ cm) with 5 cm corncob bedding and 20 glass marbles overlaid in a $4 \times 5$ equidistant arrangement. Background white noise (approximately 70 dB) was used during trials. The mouse was placed in the corner of the cage and testing consisted of a 30 min exploration period. The number of marbles that were at least two-thirds buried at the end of the trial were counted as buried.

### Elevated Plus Maze

The elevated plus maze was performed using a black, plexiglass elevated plus maze (apparatus with two open and two enclosed arms $33 \times 6$ cm with a wall of 25 cm on the closed arm, elevated 60 cm from the floor; Med Associates, St. Albans, VT). Lighting for the maze was set at 200 lux in the center of the plus maze, 270 lux on the open arms, and 120 lux on the closed arms. Background white noise (approximately 70 dB) was used during trials. Mice were placed in the center of the elevated plus maze and left to explore for 5 min in dim light condition. Mice were recorded using Ethovision XT and the time spent in the open and closed arms and the number of entries from the closed to the open arm was calculated.

### Force Swim Test

The Porsolt FST was conducted using vertical clear glass cylinder (10 cm in diameter, 25 cm in height) filled with water (25°C). The mice spent 6 min in the water and immobility was scored from 2 to 6 min. A mouse was regarded as immobile when floating motionless or making only those movements necessary to keep its head above the water.

### Tail suspension test

The tails of the mice were wrapped with tape that covered approximately 4/5 of the tail length and then fixed upside down on a hook. The immobility time of each mouse was recorded and tracked over a 6 min period using Ethovision XT.

### 2-bottle Sucrose test

Mice undergoing the sucrose preference test were habituated to 1% sucrose solution or water for 4 days and the bottle position was counterbalanced across days. Mice were individually housed in cages during the testing phase. On testing day, mice were water-deprived (12 hr) and then presented with pre-weighed identical bottles of water and 1% sucrose solution for 12 hr. The sucrose preference was calculated by dividing the volume of sucrose solution consumed by the total volume of liquid consumed.

### Rotarod

Motor coordination was assessed using an accelerating rotarod apparatus (IITC Life Science, Woodland Hills, CA). The apparatus consists of a rod covered in rubber that is approximately 30 mm in diameter and is approximately15cm above the floor of the bay. Mice performed 3 trials over 2 days. Mice were placed on a rod that accelerated from 4 to 27 rpm over 5 min and the time to fall to the

floor of the apparatus, or to turn around one full revolution while hanging onto the drum was measured.

### Locomotor activity

Locomotor activity was performed in 40 × 40×35 cm chambers (Stoelting Co, Wood Dale, IL) and distance traveled was recorded for 2 hr using Anymaze video-tracking software and analyzed in 10 min bins.

## Analysis of emotionality score

The behavioral paradigm used to calculate the emotionality score were performed in the following order with 3–4 days between testing to minimize the effects of the previous test: MB, EPM, TST and FST. To obtain a comprehensive measure for emotionality, we used z-scoring methodology to integrate standard measures of anxiety-like and depressive-like behaviors, as previously described (*Edgar et al., 2011*; *Guilloux et al., 2011*; *Lin and Sibille, 2015*). The testing parameters analyzed were as follows: marble burying (number of marble buried), elevated plus maze (time spent on open arm, number of entries into the open arm), tail suspension test (immobility) and forced swim test (immobility). For each parameter, the z-score for every individual animal was calculated using the following formula:

$$Z = \frac{X - \mu}{\sigma}$$

where X represents the individual data point, μ represents the mean of control group and σ represents the standard deviation of the control group. The emotionality score for each individual subject was first averaged within test, and then across each test to ensure equal weighting of all tests.

$$\text{Emotionality score} = \frac{Z_{MB} + Z_{EPM} + Z_{TST} + Z_{FST}}{\text{number of tests}}$$

The mean emotionality score for each group is an average of the individual within each group for each experiment.

## cAMP measurements

Prefrontal cortex tissue punches (2 mm) were flash frozen in liquid nitrogen before homogenization in 20 mM HEPES pH 8.0, 1 mM EDTA, 150 mM NaCl, 2 mM MgCl2, 1 mM DTT, and cOmplete protease inhibitor (Roche, Indianapolis, IN) followed by centrifugation at 2000 x g to clear nuclear debris. Total cAMP was determined by diluting supernatant 1:50 in 0.1 N HCl followed by quantification using a competitive cAMP enzyme immunoassay according to the manufacturer's guidelines (Direct cAMP ELISA kit, ENZO Life Sciences, Farmingdale, NY).

## Monoamine levels

Norepinephrine (NE), 3-methoxy-4-hydroxyphenylglycol (MHPG), 3,4-dihydroxyphenylacetyic acid (DOPAC), dopamine (DA), 5-hydroxyindoleacetic acid (5-HIAA), homovanillic acid (HVA) and serotonin (5-HT) in brain samples were analyzed using high performance liquid chromatography with electrochemical detection. The medial prefrontal cortex was microdissected from 200 μm coronal sections using a 300 μm punch under a dissecting microscope equipped with a freezing stage (Physitemp Instruments Inc., Clifton, NJ). The microdissected tissue was expelled into 60 μL of sodium acetate buffer (pH 5.0) containing the internal standard alpha-methyl dopamine (Merck and Co., Inc., Kenilworth, NJ) and stored at −70° C. Prior to analysis, the samples were thawed and 3 μL ascorbate oxidase (1 mg/mL, Sigma-Aldridge, St. Louis, MO) was added to each sample followed by centrifugation at 17,000 ×18 g for 10 min. The supernatant (60 μL) was removed from the samples and a Waters Alliance e2695 separations module was used to inject 50 μL of the supernatant onto a $C_{18}$4 μm NOVA-PAK radial compression column (Waters Associates, Inc. Milford, MA). The initial mobile phase (pH 4.1) was prepared using 8.6 g sodium acetate, 250 mg EDTA, 14 g citric acid, 80 mg octylsulfonic acid, and 80 mL methanol in 1 L of distilled water (monoamine standards and chemicals were purchased through Sigma-Aldridge, St. Louis, MO) and adjusted with small additions of octylsulfonic acid, glacial acetic acid and methanol to optimize the separation. Electrochemical detection

of amines was accomplished using an LC four potentiostat and glassy carbon electrode (Bioanalytical Systems, West Lafayette, IN) set at a sensitivity of 0.5 nA/V with an applied potential of +0.87 V versus an Ag/AgCl reference electrode. The pellet was solubilized in 200 µL of 0.4 N NaOH and protein content was analyzed using the Bradford method. A CSW32 data program (DataApex Ltd., Czech Republic) was used to determine monoamine concentrations in the internal standard mode using peak heights calculated from standards. Corrections were made for injection versus preparation volumes and monoamine concentrations were normalized by dividing pg monoamine by µg protein in the sample.

## Spine density analysis

Coronal sections (300 µm) were obtained in the region located between 2.10 mm and 1.54 mm rostral to the bregma. Pyramidal neurons of the layer 2/3 in the prelimbic region of the medial prefrontal cortex were injected with biocytin. Streptavidin-conjugated HRP was used to label the injected neurons using a green fluorescent substrate. Each section was then mounted on a slide and confocal images were obtained using at The Light Microscopy Facility, the Max Planck Florida Institute, using a confocal microscope (LSM 780; Carl Zeiss; Plan Apochromat 40x/1.3 Oil DIC M27) at room temperature. Image acquisition and processing were accomplished using ZEN 2011 (64 bit) software (Carl Zeiss, Oberkochen, Germany) with only minor manipulations of the images setting the fluorescence intensity in non-saturating conditions and maintaining similar parameters for each acquired image. Pictures were elaborated and spine density calculated using ImageJ and Neurolucida (MicroBrightField, Williston, VT) software. Only apical dendrites of second order were analyzed. At least three segments/neuron (>40 µm; average of 420 µm/mouse) with similar diameter (average of 1.06 ± 0.21 µm) that did not show intersections with other dendrites were considered for analysis. All measurements were performed by an experimenter blind to the mice genotype.

## Whole-cell patch clamp recordings

Coronal slices (300 µm) containing prefrontal cortex (AP +1.7–1.9) were cut in ice-cold aCSF (in mM: 124 NaCl, 2.8 KCl, 1.25 $NaH_2PO_4$, 2 $CaCl_2$, 1.25 $MgSO_4$, 26 $NaHCO_3$, 10 glucose, pH 7.5, bubbled with 95% $O_2$/5% $CO_2$) using a vibrating tissue slicer (VT1200, Leica). The slices were placed into individual wells of a custom slice incubation chamber where they remained in oxygenated aCSF at 32–36°C until use. During recording, PFC slices were transferred to a submerged recording chamber where they were continuously perfused with oxygenated aCSF with picrotoxin (100 µM), and maintained at 32–36°C. Voltage-clamp whole-cell recordings were obtained with borosilicate glass pipettes (2–5 MΩ) filled with the following solution (in mM): 115 CsMeSO₃, 10 CsCl, 5 NaCl, 10 HEPES, 0.6 EGTA, 20 TEA, 4 MgATP, 0.3 Na₂GTP, 5 QX314, with a pH of 7.35 and osmolarity of 310 mOsmol. Data were collected from neurons with resting membrane potential (RMP) more negative than −60 mV and initial series resistance ≤20 MΩ. Cells with series resistance changed more than 25% during recording period were excluded from analysis. Spontaneous excitatory postsynaptic currents (sEPSCs) were recorded at −70 mV. Evoked EPSCs were recorded using a bipolar stimulating electrode located ~100 µm away from the neurons. AMPAR- and NMDAR-mediated components were identified according to their distinct activation mechanisms and deactivation kinetics (*Chung et al., 2015*; *Myme et al., 2003*). AMPAR mediated EPSCs were recorded at −70 mV and measured as the peak response following the stimulus. NMDAR mediated EPSCs were recorded at +40 mV and measured as the mean current over a 5 ms window, 50 ms after the stimulus. Mean EPSCs were an average of 10–15 sweeps obtained at 0.1 Hz. For paired-pulse ratio (PPR) recordings, a paired-pulse protocol of two stimuli at an inter pulse interval of 50 ms was applied while the cells were voltage-clamped at −70 mV. Paired-pulse ratio was defined as the second peak amplitude (P2) divided by the first peak amplitude (P1). Mean PPRs were an average of 5–10 sweeps acquired at 1/15 Hz. Synaptic responses were collected with a HEKA EPC10 amplifier system (HEKA Instruments, Holliston, MA). Data were transferred to a PC computer using an ITC-16 digital-to-analog converter (HEKA Instruments). The signals were filtered at 2.9 kHz and digitized at 10 kHz using Patchmaster software (HEKA Instruments). Data were analyzed offline using Patchmaster. Voltages were not corrected for the liquid-liquid junction potential.

## Preparation of acute cortical slices

Acute coronal cortical slices containing medial prefrontal cortex (mPFC) were prepared from C57BL/6 wild type and $Gpr158^{-/-}$ mice, P50-60. Mice were anesthetized with isoflurane and decapitated. The brain was removed from the skull and rapidly placed in ice-cold cutting solution containing (in mM): 215 sucrose, 20 glucose, 26 $NaHCO_3$, 4 $MgCl_2$, 4 $MgSO_4$, 1.6 $NaH_2PO_4$, 1 $CaCl_2$ and 2.5 KCl. Cortical slices (400 μm thick) were prepared using a VT1000S vibrating microtome (Leica). Slices were incubated at 32°C for 30 min in a holding chamber containing 50% cutting solution and 50% artificial cerebrospinal fluid (ACSF; in mM: 127 NaCl, 25 $NaHCO_3$, 1.25 $NaH_2PO_4$, 2.5 KCl, 25 D-glucose, 2 $CaCl_2$, and 1 $MgCl_2$). After 30 min, this solution was replaced with ACSF at room temperature. Slices were allowed to recover for more than 1 hr in ACSF before recording. All solutions were equilibrated for at least 30 min with 95%$O_2$/5%$CO_2$.

## Two-photon imaging, electrophysiology, and glutamate uncaging

Layer 2/3 pyramidal neurons of prelimbic (PrL) subregion in the mPFC at depths of 20–40 μm were imaged using a two-photon microscope (Prairie Technologies, Inc) with a pulsed Ti::sapphire laser (MaiTai HP DeepSee, Spectra Physics) tuned to 920 nm (2–2.5 mW at the sample) in recirculating ACSF aerated with 95%$O_2$/5%$CO_2$ containing (in mM): 2 $Ca^{2+}$, 1 $Mg^{2+}$, 0.001 TTX, and 2.5 MNI-caged-glutamate. For each neuron, image stacks (512 × 512 pixels; 0.035 μm / pixel) with 1 μm z-steps were collected from one segment of secondary or tertiary apical and/or basal dendrites 30–80 μm from the soma. All images shown are maximum projections of 3D image stacks after applying a median filter (2 × 2) to the raw image data. Whole-cell recordings (electrode resistances 5–8 MΩ; series resistances 20–40 MΩ) were performed at 25°C on visually identified PrL layer 2/3 pyramidal neurons within 40 μm of the slice surface using a MultiClamp 700B amplifier (Molecular Devices, Sunnyvale, CA). In brief, to record uncaging-evoked excitatory postsynaptic currents (uEPSCs), PrL neurons were patched in voltage-clamp configuration (Vhold = −65 mV and +40 mV for AMPA receptor- and NMDA receptor-mediated uEPSCs, respectively) using cesium-based internal solution (in mM: 135 Cs-methanesulfonate, 10 HEPES, 10 Na2 phosphocreatine, 4 MgCl2, 4 Na2-ATP, 0.4 Na-GTP, 3 Na L-ascorbate, 0.2 Alexa 488,~300 mOsm, ~pH 7.25) in ACSF. uEPSC amplitudes from individual spines were quantified as the average (8–10 test pulses of 1 ms duration at 0.1 Hz) from a 2 ms window centered on the maximum current amplitude after uncaging pulse delivery for AMPA currents and from a 10 ms window between 70 and 80 ms after stimulus for NMDA currents. Laser pulses were delivered by parking the beam at a point ~0.5 μm from the center of the spine head with a pulsed Ti::sapphire laser (MaiTai HP, Spectra-Physics) tuned to 720 nm (16–18 mW at the sample). Integrated green (Alexa 488) fluorescence intensities were measured from background-subtracted green fluorescence using the integrated pixel intensity of a boxed region surrounding the spine head. Estimated spine size was calculated by normalizing the fluorescence intensities (as described above) for each individual spine to the mean fluorescence intensities measured from four ROIs on the dendritic shaft.

## Human subjects

Post-mortem tissues were obtained from the NIH NeuroBioBank at the University of Pittsburg, University of Miami Brain Endowment Bank and the Human Brain and Spinal Fluid Resource Center (Los Angeles). Informed consent was obtained from the next of kin for all subjects, and all procedures were approved by each institutions' internal review board; University of Pittsburgh Institutional Review Board for Biomedical Research, University of Miami Institutional Review Board and the Institutional Review Board for the Department of Veterans Affairs. Tissue from the dorsolateral prefrontal cortex from nine individuals that met the DMS-IV criteria for MDD and eight control individuals with no history of neurological or psychiatric illnesses were procured.

## Quantification and statistical analysis

Statistical analysis was performed using GraphPad Prism (Prism6.0, GraphPad, San Diego, California). Student's *t*-test was used to compare means between two groups, and one-way or two-way analysis of variance followed by Tukey's or Bonferroni *post hoc* tests were used to determine significant differences among multiple groups. Frequency distributions were compared using the Kolmogorov–Smirnov test. Statistical tests were performed two-sided unless stated otherwise. Differences

were considered significant if p<0.05. All data are expressed as means ± s.e.m. No statistical methods were used to predetermine sample sizes, but our sample sizes are similar to those generally employed in comparable studies.

## Acknowledgements

We wish to thank Ms. Natalia Martemyanova for producing and maintaining mice examined in this study, Dr. Pablo Martinez for the help with the mass-spectrometry experiments and Dr. Massimiliano Aceti for his help with the analysis of spine density. This work was supported by NIH grants, MH105482 (KAM), HL105550 (KAM), DA019921 (KJR), MH107460 (H-BK), DC014093 (SMY), the University of Iowa and the Max Planck Society (SMY) and by the Canadian Institutes of Health Research Fellowship (LPS).

## Additional information

### Funding

| Funder | Grant reference number | Author |
|---|---|---|
| National Institute of Mental Health | MH105482 | Kirill A Martemyanov |
| National Heart, Lung, and Blood Institute | HL105550 | Kirill A Martemyanov |
| National Institute of Mental Health | MH107460 | Hyungbae Kwon |
| National Institute on Drug Abuse | DA019921 | Kenneth J Renner |
| National Institute on Deafness and Other Communication Disorders | DC014093 | Samuel M Young Jr |
| University of Iowa | | Samuel M Young Jr |
| Max-Planck-Gesellschaft | | Samuel M Young Jr |
| Canadian Institutes of Health Research | | Laurie P Sutton |

The funders had no role in study design, data collection and interpretation, or the decision to submit the work for publication.

### Author contributions

Laurie P Sutton, Cesare Orlandi, Conceptualization, Data curation, Formal analysis, Investigation, Writing—original draft, Writing—review and editing; Chenghui Song, Won Chan Oh, Brian S Muntean, Xiangyang Xie, Data curation, Formal analysis, Investigation; Keqiang Xie, Alice Filippini, Resources, Investigation, Methodology; Rachel Satterfield, Resources, Methodology; Jazmine D W Yaeger, Resources, Formal analysis, Investigation; Kenneth J Renner, Data curation, Formal analysis, Investigation, Writing—review and editing; Samuel M Young Jr, Resources, Supervision; Baoji Xu, Formal analysis, Supervision, Writing—review and editing; Hyungbae Kwon, Data curation, Formal analysis, Supervision, Writing—review and editing; Kirill A Martemyanov, Conceptualization, Formal analysis, Supervision, Funding acquisition, Writing—original draft, Project administration, Writing—review and editing

### Author ORCIDs

Laurie P Sutton  http://orcid.org/0000-0002-8694-2060
Chenghui Song  http://orcid.org/0000-0003-3289-809X
Samuel M Young Jr  https://orcid.org/0000-0002-7589-7612
Kirill A Martemyanov  http://orcid.org/0000-0002-9925-7599

## Ethics

Animal experimentation: This study was performed in strict accordance with the recommendations in the Guide for the Care and Use of Laboratory Animals of the National Institutes of Health. All procedures were approved by the Institutional Animal Care and Use Committee (IACUC) protocol (#16-032) at The Scripps Research Institute.

## Decision letter and Author response

Decision letter https://doi.org/10.7554/eLife.33273.020
Author response https://doi.org/10.7554/eLife.33273.021

## Additional files

### Supplementary files

• Supplementary file 1. G protein Coupled Receptors identified by mass-spectrometry in the PFC.
DOI: https://doi.org/10.7554/eLife.33273.017

• Transparent reporting form
DOI: https://doi.org/10.7554/eLife.33273.018

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
