## [Decision Letter]

Thank you for submitting your article "Orphan receptor GPR158 controls stress-induced depression" for consideration by *eLife*. Your article has been favorably evaluated by a Senior Editor and three reviewers, one of whom is a member of our Board of Reviewing Editors. The following individual involved in review of your submission has agreed to reveal his identity: Thomas L. Kash (Reviewer #2).

The reviewers have discussed the reviews with one another and the Reviewing Editor has drafted this letter to help you prepare a revised manuscript.

This paper proposes that the abundance of an orphan G-protein coupled receptor (Gpr158) in the pre-frontal cortex (PFC) increases with stress (or glucocorticoid treatment) and that promotes depression-like behaviors of mice. Deliberate overexpression of GPR158 (by viral transduction) in PFC promotes depression-like behaviors and knockout of Gpr158 gene (everywhere) has anti-depression effects.

All three reviewers agree that this is an intriguing study that begins to assign a function to Gpr158; however, there are some major concerns that need to be addressed prior to publication.

Major points:

1) It is important to delineate the functions of Gpr158 in specific populations of PFC neurons because the functions in glutamatergic and GABAergic neurons may differ. Without that information, overexpression of Gpr158 in all PFC neurons is difficult to interpret.

2) Of the many GABAergic neurons, is Gpr158 expression restricted to the parvalbumin-expressing neurons?

3) The signaling pathways engaged in glutamatergic and GABAergic neurons should be established, at least to the level of knowing whether it is excitatory or inhibitory.

4) Because Gpr158 is expressed widely in the brain, the effect of knockout or knockdown of Gpr158 in the PFC (ideally in pyramidal or glia cells) on behavior is needed.

5) The authors should discuss the possibility that osteocalcin is an endogenous ligand for Gpr158 and what significance that has for interpretation of their results.

6) The authors should validate the antibody used in their studies more thoroughly and describe the genetic background and breeding strategy.

*Reviewer #1:*

The GPR158-Ad virus presumably infects all cells in the PFC, including those that do not normally express this GPCR. Thus, it is not possible to conclude that upregulation of GPR158 in cells where it is normally expressed promotes depression-like behaviors, it only shows that this ectopic expression paradigm seems to work. There is no evidence that the PFC is a specific region where GPR158 can affect depression-like behaviors-perhaps increasing GPR158 levels in other brain regions is also effective.

The observation that GPR158 is in both excitatory pyramidal neurons and GABAergic PV neurons in the PFC raises the question of whether this GPCR signaling is more important and whether the signaling mechanism is the same in both cell types.

The behavioral studies with GPR158 KO and WT mice are nice, but they do not bear on the brain region where this orphan receptor is important. Restoring GPR158 in neurons that normally make it in the PFC on a KO background and performing the behavioral and electrophysiological experiments would go a long way towards solidifying this story.

The genetic background of the GPR158 WT and KO mice used here should be indicated. Were the WT and KO mice used for these studies littermates derived from breeding heterozygotes? If not, there is concern that WT and KO mice used for studies may have subtle genetic background differences.

A recent study suggests that GPR158 is activated by osteocalcin (J Exp Med 214, 2859), that it participates in hippocampal memory and that it is coupled to G-α-q signaling. The results in this study do not seem to be compatible with G-α-q signaling.

*Reviewer #2:*

This is a novel finding, identifying GPR158 as an important mediator of stress induced depression like behavior. The authors test this idea by over expressing and knocking out this receptor and assessing behavior. In addition, the authors provide data showing changes of this GPCR in human MDD patient brain. This is a nice addition that lends cross species generalizability. However, some of the experiments are concerning:

1) Over-expression is not directed to any cell type, so while the behavioral data is robust, it is challenging to interpret. Of additional concern, this is only explored in male mice. given later data in the KO mice was examined in males and female mice, seem like a useful comparison.

2) The knockout data is compelling in such that deletion from birth can induce changes in behavior, synaptic function independent of stress exposure. This raises the issue that there could be a developmental compensation from the deletion of the receptor.

Because of these issues, I think a more robust analysis would be a viral knockdown of GPR158 prior to stress exposure to parse out role of developmental deletion. These issues are not addressed in the manuscript, and given the novelty of these findings and the potential length of time it would take to repeat these experiments with a knockdown approach, I would also be favorable if there were a more in depth discussion of these concerns in the manuscript.

3) The authors present data showing that layer 2/3 neurons in the mPFC are altered. Given the expression of this protein in nearly all GABA neurons, this is an incomplete analysis. These particular experiments, recording from different cell types in KO and WT mice, would be very helpful in clarifying molecular/synaptic adaptations.

4) The near total exclusion of this GPCR in SST neurons is stunning. It is a bit confusing how the expression in GABA neurons is so high, but nothing in SST. It would be useful to flush this out a bit more with triple labeling, to provide further evidence of this selective expression.

*Reviewer #3:*

This is a very interesting manuscript from Martemyanov and colleagues showing that expression of the orphan GPCR GPR158 is glucocorticoid-regulated, is highly expressed in rodent prefrontal cortex, and is upregulated in post-mortem dlPFC from depressed human patients and in rodents subjected to behavioral procedures that precipitate depression-like behavioral responses. Furthermore, virus-mediated GPR158 overexpression in PFC precipitates depression-like behaviors in rodents and that knockout mice show the opposite.

Overall, this is a rigorously-performed and well-written study that is likely to be of considerable interest to the field, particularly those interested in new medications development.

I have only a few comments for the authors' consideration:

1) It is important to validate the antibody used for immunofluorescence work in the KO mice. If it is selective, then the signal should not be detected in the PFC (or elsewhere) in the KO mice. It is understood that bands were not detected in KO tissues when using this antibody for immunoblotting. But, frequently antibodies that are relatively selective when used for immunoblotting will bind nonspecifically when used for immunoimaging.

2) It appears from the presented immunoimaging that GPR158 is not cortically-enriched but rather expressed in most brain regions captured in the representative brain slice that is shown, with the exception of the white matter tracts as expected. This begs the question of whether GPR158 is active in other brain regions and if similar effects as those reported for the PFC in mice would have been detected when GPR158 expression is manipulated in other brain regions implicated in the regulation of depression-related behaviors, such as the hippocampus, accumbens etc. In other words, there are no anatomical controls in the current data-set to confirm that reported effects are anatomically selective. This is not a major issue, as the reported data are compelling, but some data related to other brain sites would have further strengthened the manuscript.

3) Greater consideration should be given to the endogenous agonist for GPR158, and more broadly the likely signaling mechanisms for the receptor should be presented in the manuscript. I am not suggesting that the authors seek to de-orphanize the receptor as this is beyond the scope of this manuscript. However, is there any evidence from the published literature that hints at the identity of the endogenous agonist? Is GPR158 directly activated by glucocorticoids? To which intracellular signaling complex does GPR158 couple?

---

## [Author Response]

Major points:1) It is important to delineate the functions of Gpr158 in specific populations of PFC neurons because the functions in glutamatergic and GABAergic neurons may differ. Without that information, overexpression of Gpr158 in all PFC neurons is difficult to interpret.

The current manuscript deals primarily with the role of Gpr158 in glutamatergic pyramidal neurons in the PFC. The following lines of evidence that we present indicates that Gpr158 significantly impacts their function. 1) morphological changes induced by Gpr158 loss occur in glutamatergic neurons (Figure 7); 2) changes in synaptic strength and AMPA receptor function upon ablation of Gpr158 also happen in glutamatergic neurons (Figure 7 and Figure 7—figure supplement 1). The identity of neurons examined in these datasets may not have been made sufficiently clear and we clarified this upon revision. Additionally, we solidify the role of Gpr158 in glutamatergic neurons by supplying the following new data in the revised manuscript. 3) We showed that stress specifically regulates Gpr158 levels only in glutamatergic and not GABAergic neurons (Figure 2). In these experiments we subjected mice to stress and examined cell-specificity of Gpr158 modulation. Because Gpr158 plays a role in controlling depressive–like phenotypes that we document by both gain and loss of function studies and because these changes are precipitated by stress we think that these data serve as an additional evidence for the role of Gpr158 in glutamatergic neurons in controlling behavioral outcomes. 4) We additionally report changes in processing neurotransmitter inputs onto glutamatergic neurons obtained by patch clamping glutamatergic neurons specifically and recording their sEPSC (Figure 7 and Figure 1—figure supplement 1). The modulation of sEPSC frequency that we observe in this population further argues for the involvement of Gpr158 in controlling the signal processing by glutamatergic neurons. We feel that together this dataset provides compelling evidence pointing to Gpr158 action in glutamatergic neurons.

2) Of the many GABAergic neurons, is Gpr158 expression restricted to the parvalbumin-expressing neurons?

The format and quantification that we used in the original manuscript may have been unclear and did not directly support the intent of examining the cell specificity of Gpr158 expression. In our study we used latest generation of in situ hybridization technique that provides cellular resolution of mRNA expression to define the expression of Gpr158 in the cellular populations of the PFC based on double labeling with known cellular markers. In accordance with the published literature we detect that ~70% in the region are glutamatergic, ~11% are GABAergic and ~19% are glia. The expression of Gpr158 is detected in all glutamatergic neurons and none in glia. We also detect Gpr158 expression in most GABAergic neurons. However, given their diversity, the attribution to specific subtypes is not as straightforward due to overlapping marker expression, lack of the specific subtype markers for many Gpr158-expressing cells and limitation of the in situ technique that only allows dual labeling. On re-examination of the data we came to the conclusion that we cannot confidently determine the identity of the SST+ cells that do not express Gpr158. In the revised manuscript we simplified the analysis to avoid possible confusion and misattribution by focusing on Gpr158 expressing cells and defining what markers they express. This analysis is straightforward and shows that 90% of Gpr158 positive cells are glutamatergic neurons, 10% are Gad1 positive, 1.5% are parvalbumin positive and none of the ~750 cells examined in each double in situ hybridization experiment show co-expression with glial markers or somatostatin (Figure 1—figure supplement 2). We feel that because the current study emphasized the role of Gpr158 in glutamatergic neurons and does not study the effects in GABAergic population, subdivision of their expression in GABAergic neurons may be not essential but rather detracting from the main points of the paper, hence these claims were de-emphasized.

3) The signaling pathways engaged in glutamatergic and GABAergic neurons should be established, at least to the level of knowing whether it is excitatory or inhibitory.

In the revised manuscript we provide the results of systematic analysis of signaling pathways engaged by Gpr158 (Figure 8). We performed a screen looking at a variety of signaling molecules impacted by Gpr158 ablation in the PFC (Figure 8). Most notably this includes the discovery that Gpr158 loss significantly upregulates neurotrophic factor BDNF and that this effect is mediated by changes in eEF2 phosphorylation which regulates local translation rather than affected by transcriptional upregulation (Figure 8). Identification of BDNF as a signaling mediator of Gpr158 effects is exciting because it further mechanistically links signaling changes to observed changes in neuronal morphology as well as antidepressant effects that alter AMPA/NMDAR function, such as ketamine. Given consistent increase in virtually all measures reporting increase in neuronal activity: synaptic strength, sEPSC), spine number, cAMP, AMPA receptor currents and BDNF expression observed upon Gpr158 loss, we think this data strongly suggests that Gpr158 is an inhibitory receptor. This point is addressed in the revised manuscript.

4) Because Gpr158 is expressed widely in the brain, the effect of knockout or knockdown of Gpr158 in the PFC (ideally in pyramidal or glia cells) on behavior is needed.

The main emphasis of this paper is on molecular mediators that control stress induced changes in depression. We identified that stress upregulates Gpr158 levels specifically in the PFC. To further reinforce regional specificity of this effect we added results examining modulation of Gpr158 levels across different brain regions involved in depressive like behaviors (Cerebellum, Caudate/Putamen, Nucleus Accumbens and Hippocampus) that shows exclusive modulation of Gpr158 in the PFC prompting us to focus current study on this region (Figure 1—figure supplement 1). The viral overexpression of Gpr158 in the PFC and the rescue of Gpr158 specifically in the PFC of knockout mice provides sufficient evidence supporting the contribution of Gpr158 in this region to stress mediated depression. Given that we report that Gpr158 upregulates AMPA receptor function in the PFC, we solidify our evidence by demonstrating that partial blockade of AMPA receptor function by subthreshold doses on NBQX administration reverses the antidepressant-like effects in Gpr158 knockout mice (Figure 7). Lastly, given the lack of Gpr158 expression in glia, the neuronal specific tropism of Ad virus with synaptophysin promoter that we used for the expression of Gpr158 in the PFC, we do not think that glial specific knockdown or rescue experiments are warranted. Similarly, as discussed above, the lack of Gpr158 regulation by stress in GABAergic neurons makes knockdown or rescue of this protein specifically in this population tangential to the main emphasis and conclusions of this study.

We would also like to note that although we show changes in PFC to be sufficient in explaining Gpr158 behavioral effects (viral overexpression/knockout in the PFC) we cannot formally exclude possible role that Gpr158 also plays in other regions in contributing to anti-depressant behaviors (unrelated to stress), which in no way invalidates our findings in the PFC. Collectively, we are able to conclude that Gpr158 in the glutamatergic neurons in the mPFC is sufficient to drive stress-induced changes in the behavior. This will not completely exclude the role Gpr158 has in GABAergic neurons but we believe this line of research is beyond of the scope of this paper.

5) The authors should discuss the possibility that osteocalcin is an endogenous ligand for Gpr158 and what significance that has for interpretation of their results.

We have addressed this in the Discussion section of the revised manuscript. The rapid progress in understanding GPR158 functions in the brain is exciting. We would like to note that the osteocalcin report was published while this manuscript was already in the peer review process. We are certainly happy to discuss this new development, however, we feel that extending our report along these lines experimentally may better be suited for future studies. Comparing our results in PFC with the data reported in the hippocampus reveals that there may be major significant differences in GPR158 signaling properties between neurons of PFC and hippocampus. While this is not entirely unusual for a GPCR to exhibit region selective signaling bias, determining the exact mechanisms and possible presence of additional endogenous ligands is outside of the scope of the current report, which is already quite extensive.

6) The authors should validate the antibody used in their studies more thoroughly and describe the genetic background and breeding strategy.

The antibodies used were for Western blotting and have been thoroughly validated using KO tissues. The results showed the complete lack of the band that we consider to be Gpr158, thus confirming the specificity for this application. We did not use these antibodies for the immunohistochemistry because they produce diffuse neuropilar staining pattern (although specific), which does not allow easy attribution of cell specificity. Therefore, instead we have used the latest generation fluorescent in situ hybridization that provides sensitive detection of mRNA expression with high specificity and cellular resolution. We may not have been sufficiently clear as to what technique we have used to characterize Gpr158 expression and we have since clarified this in the revised manuscript. In addition, we have included control that established specificity of in situ hybridization reactions (Figure 1—figure supplement 2).

In terms of the genetic background – the mice were generated on a pure C57BL6 background and have been maintained as heterozygous breeding pairs producing -/- and +/+ littermates used of all of the studies in the paper. We relied exclusively on littermates for all the comparisons. Again, we will clarify this in the revised manuscript.

Reviewer #1:The GPR158-Ad virus presumably infects all cells in the PFC, including those that do not normally express this GPCR. Thus, it is not possible to conclude that upregulation of GPR158 in cells where it is normally expressed promotes depression-like behaviors, it only shows that this ectopic expression paradigm seems to work. There is no evidence that the PFC is a specific region where GPR158 can affect depression-like behaviors-perhaps increasing GPR158 levels in other brain regions is also effective.

To address regional specificity we have analyzed Gpr158 protein levels following stress in other brain regions including nucleus accumbens and hippocampus (Figure 1—figure supplement 1). However, we have only observed the upregulation of Gpr158 in the mPFC following chronic-stress exposure. While we cannot rule out the importance of Gpr158 in other brain regions, we can conclude that Gpr158 in the mPFC is sufficient to drive stress-mediated responses. We have also addressed this concept of regional specificity in the Discussion.

The observation that GPR158 is in both excitatory pyramidal neurons and GABAergic PV neurons in the PFC raises the question of whether this GPCR signaling is more important and whether the signaling mechanism is the same in both cell types.The behavioral studies with GPR158 KO and WT mice are nice, but they do not bear on the brain region where this orphan receptor is important. Restoring GPR158 in neurons that normally make it in the PFC on a KO background and performing the behavioral and electrophysiological experiments would go a long way towards solidifying this story.

The current paper focuses on the role of Gpr158 in glutamatergic neurons. Here we have shown that the morphological changes induced by Gpr158 loss occur in glutamatergic neurons (Figure 7) and the changes in synaptic strength and AMPA receptor function in Gpr158 KO also happen in glutamatergic neurons (Figure 7 and Figure 7—figure supplement 1). The identity of neurons examined in these experiments may not have been made clear and we have clarified this in the revision. Furthermore, we have added additional data to demonstrate the role of Gpr158 in glutamatergic neurons. In series of experiments we examined the cell-specific modulation of Gpr158 levels following chronic stress. We found that the increase in Gpr158 levels were only found in glutamatergic and not GABAergic neurons (Figure 2). We also investigated the changes in processing neurotransmitter inputs onto glutamatergic neurons obtained by patch clamping glutamatergic neurons specifically and recording their sEPSC (Figure 7). Together, this demonstrates the role of Gpr158 action in glutamatergic neurons. We do not argue that Gpr158 in the GABAergic neurons may have an additional role in modulating behavioral outcomes but it is beyond the scope of this paper to examine this contribute in depth.

The genetic background of the GPR158 WT and KO mice used here should be indicated. Were the WT and KO mice used for these studies littermates derived from breeding heterozygotes? If not, there is concern that WT and KO mice used for studies may have subtle genetic background differences.

The breeding strategy and genetic background has been clarified in the Materials and methods section. The mice were generated on a C57/BL6 background and were maintained as heterozygous breeding pairs to generate Gpr158^-/-^ and Gpr158^+/+^ mice that were used in the studies. We relied exclusively on littermates for all the comparisons.

A recent study suggests that GPR158 is activated by osteocalcin (J Exp Med 214, 2859), that it participates in hippocampal memory and that it is coupled to G-α-q signaling. The results in this study do not seem to be compatible with G-α-q signaling.

Please see our response to major point #5.

Reviewer #2:1) Over-expression is not directed to any cell type, so while the behavioral data is robust, it is challenging to interpret. Of additional concern, this is only explored in male mice. given later data in the KO mice was examined in males and female mice, seem like a useful comparison.

The reviewer is correct in that the over-expression of Gpr158 from the viral manipulations is not specific to any cell type. However, to address the concerns of cell-type specificity we have identified in which cell-type Gpr158 is affected following chronic stress. In these experiments mice were subjected to chronic stress and using in situ hybridization techniques we were able to quantify Gpr158 mRNA levels in glutamatergic and GABAergic neurons. We found that stress specifically regulates Gpr158 levels only in glutamatergic and not GABAergic neurons. This additional data can be found in Figure 2. We found that both male and female knockout mice showed the same phenotype in the behavioral assays (marble burying, elevated plus maze, tail suspension and force swim test) and as such the subsequent viral overexpression studies were only performed in male mice.

2) The knockout data is compelling in such that deletion from birth can induce changes in behavior, synaptic function independent of stress exposure. This raises the issue that there could be a developmental compensation from the deletion of the receptor.Because of these issues, I think a more robust analysis would be a viral knockdown of GPR158 prior to stress exposure to parse out role of developmental deletion. These issues are not addressed in the manuscript, and given the novelty of these findings and the potential length of time it would take to repeat these experiments with a knockdown approach, I would also be favorable if there were a more in depth discussion of these concerns in the manuscript.

Like all studies involving transgenic mice there is always a concern that the deletion of a specific protein maybe due to a developmental compensation. However, the viral studies whereby we overexpressed Gpr158 in mPFC of adult mice (Figure 4) and rescued Gpr158 in the knockout model (Figure 5—figure supplement 2) of adult mice suggests that the behavioral effects are not due to developmental compensation from the lack of Gpr158. We have also addressed this issue in the Discussion.

3) The authors present data showing that layer 2/3 neurons in the mPFC are altered. Given the expression of this protein in nearly all GABA neurons, this is an incomplete analysis. These particular experiments, recording from different cell types in KO and WT mice, would be very helpful in clarifying molecular/synaptic adaptations.

As suggested by the reviewer we have performed additional experiments directly recording from glutamatergic neurons in layer of 2/3 PFC. The results additionally corroborated our conclusions about alterations in the function of these neurons caused by GPR158 deletion and were incorporated in the revised manuscript (Figure 7). For the reasons explained above (see responses to major points #1 and #2) we restricted our focus on studying glutamatergic neurons only.

4) The near total exclusion of this GPCR in SST neurons is stunning. It is a bit confusing how the expression in GABA neurons is so high, but nothing in SST. It would be useful to flush this out a bit more with triple labeling, to provide further evidence of this selective expression.

We have addressed this concern in our response to the major point #2.

Reviewer #3:1) It is important to validate the antibody used for immunofluorescence work in the KO mice. If it is selective, then the signal should not be detected in the PFC (or elsewhere) in the KO mice. It is understood that bands were not detected in KO tissues when using this antibody for immunoblotting. But, frequently antibodies that are relatively selective when used for immunoblotting will bind nonspecifically when used for immunoimaging.

The antibodies used were for Western blotting and have been thoroughly validated using KO tissues. The results showed the complete lack of the band that we consider to be Gpr158, thus confirming the specificity for this application. We did not use these antibodies for the immunohistochemistry because they produce diffuse neuropilar staining pattern (although specific) which does not allow easy attribution of cell specificity. Therefore, instead we have used the latest generation fluorescent in situ hybridization that provides sensitive detection of mRNA expression with high specificity and cellular resolution. The current manuscript may have not been sufficiently clear as to what technique we have used to characterize Gpr158 expression and we have since clarified this issue. In addition, we included control that establishes specificity of in situ hybridization reactions (Figure 1—figure supplement 2).

2) It appears from the presented immunoimaging that GPR158 is not cortically-enriched but rather expressed in most brain regions captured in the representative brain slice that is shown, with the exception of the white matter tracts as expected. This begs the question of whether GPR158 is active in other brain regions and if similar effects as those reported for the PFC in mice would have been detected when GPR158 expression is manipulated in other brain regions implicated in the regulation of depression-related behaviors, such as the hippocampus, accumbens etc. In other words, there are no anatomical controls in the current data-set to confirm that reported effects are anatomically selective. This is not a major issue, as the reported data are compelling, but some data related to other brain sites would have further strengthened the manuscript.

Gpr158 is indeed broadly expressed in the brain. To address regional specificity, we have analyzed Gpr158 protein levels following stress in other brain regions including nucleus accumbens, caudate/putamen, cerebellum and hippocampus (Figure 1—figure supplement 1). We only observed the upregulation of Gpr158 in the mPFC following chronic-stress exposure. While we cannot rule out the importance of Gpr158 in other brain regions, we can conclude that Gpr158 in the mPFC is sufficient to drive stress-mediated responses. We have also addressed this concept of regional specificity in the Discussion.

3) Greater consideration should be given to the endogenous agonist for GPR158, and more broadly the likely signaling mechanisms for the receptor should be presented in the manuscript. I am not suggesting that the authors seek to de-orphanize the receptor as this is beyond the scope of this manuscript. However, is there any evidence from the published literature that hints at the identity of the endogenous agonist? Is GPR158 directly activated by glucocorticoids? To which intracellular signaling complex does GPR158 couple?

In the revised manuscript we provide evidence for signaling pathways engaged by Gpr158. We performed a screen looking at a variety of signaling molecules impacted by Gpr158 ablation in the PFC (Figure 8). Most notably this includes the discovery that Gpr158 loss significantly upregulates neurotrophic factor BDNF and that this effect is mediated by changes in eEF2 phosphorylation which regulates local translation rather than affected by transcriptional upregulation (Figure 8). Identification of BDNF as a signaling mediator of Gpr158 effects is exciting because it further mechanistically links signaling changes to observed changes in neuronal morphology as well as antidepressant effects that alter AMPA/NMDAR function, such as ketamine. We have also discussed possible endogenous agonists for Gpr158 and its possible region selective effects on signaling in the Discussion section.